# Feature-based comparison of sea-ice deformation in lead-permitting sea-ice simulations

Nils Hutter[1] and Martin Losch[1]

[1]Alfred-Wegener-Institut für Polar-und Meeresforschung, Bremerhaven, Germany.

**Correspondence:** Nils Hutter (nils.hutter@awi.de)

**Abstract.** The sea-ice modelling community progresses towards Pan-Arctic simulations that explicitly resolve leads in the simulated sea-ice cover. Evaluating these simulations against observations poses new challenges. A new feature-based evaluation of simulated deformation fields is introduced and the results are compared to a scaling analysis of sea ice deformation. Leads and pressure ridges – here combined into Linear Kinematic Features (LKF) – are detected and tracked automatically from deformation and drift data. LKFs in two Pan-Arctic sea-ice simulations with a horizontal grid spacing of 2 km are compared with an LKF data-set derived from the RADARSAT Geophysical Processor System (RGPS). One simulation uses a 5-class Ice Thickness Distribution (ITD). The simulated sea-ice deformation follows a multi-fractal spatial and temporal scaling as observed from RGPS. The heavy-tailed distribution of LKF lengths and the scale invariance of LKF curvature, which points to the self-similar nature of sea-ice deformation fields, is reproduced by the model. Interannual and seasonal variations of the number of LKFs, LKF densities, and LKF orientations in the ITD simulation are found to be consistent with RGPS observations. The lifetimes and growth rates follow a distribution with an exponential tail. The model overestimates the intersection angle of LKFs, which is attributed to the model's viscous-plastic rheology with an elliptical yield curve. In conclusion, the new feature-based analysis of LKF statistics is found to be useful for a comprehensive evaluation of simulated deformation features, which is required before the simulated features can be used with confidence in the context of climate studies. As such it complements the commonly used scaling analysis and provides new useful information for comparing deformation statistics. The ITD simulation is shown to reproduce LKFs sufficiently well to be used for studying the effect of directly resolved leads in climate simulations. The feature-based analysis of LKFs also identifies specific model deficits that may be addressed by specific parameterizations, for example, a damage parameter, a grounding scheme, and a Mohr-Coulombic yield curve.

## 1 Introduction

Current efforts in the sea-ice modeling community push sea-ice models to Pan-Arctic lead-permitting sea-ice simulations. In these simulations, the Arctic ice cover consists of individual "floes" that are formed by strongly localized deformation along the emerging floe boundaries. There are two approaches to obtain such a behaviour: (1) a very fine grid-spacing ($< 5$ km) and the classic viscous-plastic (VP) rheology (Hutter et al., 2018; Wang et al., 2016; Spreen et al., 2016) or (2) new rheological frameworks (e.g. Maxwell elasto-brittle (MEB), Dansereau et al., 2016). The emergence of deformation features, which can be identified as leads and pressure ridges, calls for a proper evaluation of model simulations against observations. This is

challenging because ice mechanics are non-linear and chaotic. A direct comparison of deformation fields bears similar issues as comparing eddy resolving ocean model simulations to high-resolution satellite observations (Mourre et al., 2018). Therefore, it should not be attempted if accurate initial conditions (e.g. obtained by data assimilation) are not available.

Resolving leads in sea-ice simulations opens up new possibilities in Arctic climate research and sea-ice forecasting. Leads are openings in the sea-ice cover where direct atmosphere-ocean processes are strong. A sea-ice component including leads allows for the direct simulation of these interactions in regional or global climate models. The distribution of leads also has a strong impact on the local drift field. Reliable short-term sea-ice drift forecasts will therefore depend on the model's capacity to simulate and initialize fields with localized deformation. The increasing economic interest in the Arctic (e.g. shipping, Eguíluz et al., 2016) requires skillful predictions in these remote regions (Jung et al., 2016) to prevent environmental catastrophes and to organize search and rescue operations. A realistic representation of deformation features in sea-ice models is the prerequisite of both applications.

In the past, high-resolution sea-ice simulation were mostly evaluated with respect to their simulated deformation fields. Here, calculating scaling characteristics of sea-ice deformation was the most common method (Girard et al., 2009; Rampal et al., 2016; Spreen et al., 2016; Bouchat and Tremblay, 2017; Hutter et al., 2018). The scaling statistics make use of the observed power-law scaling of sea-ice deformation (Marsan et al., 2004) and determine the degree of heterogeneity and intermittency of sea-ice deformation for satellite observations and model simulations (Rampal et al., 2016; Hutter et al., 2018). The underlying idea of these metrics is that the presence of extreme values and strong localization in sea-ice deformation indicates a realistic representation of deformation features.

While scaling characteristics give some insight into the underlying material properties of sea ice, their interpretation with respect to individual deformation features is not straightforward (Bouchat and Tremblay, 2017; Hutter et al., 2018). Various metrics for evaluating LKFs as discontinuities in the deformation fields have been suggested, but they all provide only a summary of agreement with a reference in a single score for the entire LKF field (Coon et al., 2007; Levy et al., 2008; Mohammadi-Aragh et al., 2018). A comprehensive description of individual deformation features requires their detection to extract statistics such as density, orientation, intersection angle, and persistence. A new LKF detection and tracking algorithm (Hutter et al., 2019) identifies single LKFs in deformation data both derived from satellite observations and simulated by sea-ice models. The resulting data set provides ample opportunity to compare various spatial characteristics and the temporal evolution of LKFs.

The objective of this paper is to establish a feature-based evaluation of sea-ice deformation in lead-permitting sea ice simulations. We apply the LKF detection and tracking algorithm of Hutter et al. (2019) to two different sea-ice simulations with a horizontal grid-spacing of 2 km, one of which uses an Ice Thickness Distribution (ITD). We compare the extracted LKFs to an LKF data-set derived from RADARSAT Geophysical Processor System (RGPS) deformation data (Hutter et al., 2019) with respect to their Pan-Arctic distribution (density and orientation), their spatial properties (length, curvature, and intersection angle), and their temporal characteristics (persistence and growth rates). In addition, we test which conclusions about the properties of LKFs can be drawn from a spatio-temporal scaling analysis of sea ice deformation (following, e.g. Rampal et al., 2016; Hutter et al., 2018). By analyzing two different model simulations, we study how changes to the model physics, in our

case the explicit ridging processes in an ITD model, affect the simulated LKFs, and how the different analysis methods outline that difference. With our analysis we test whether the ice strength parameterization of the ITD model, which mainly depends on the thinner ice classes, accelerates lead formation by a faster feedback between deformation, ice thickness, and ice strength as suggested in (Hutter et al., 2018).

## 2 Methods

### 2.1 LKF detection and tracking algorithms

Our LKF detection and tracking algorithms (Hutter et al., 2019) splits the detection of LKFs in sea-ice deformation fields into three steps: (i) the algorithm classifies pixels with locally higher deformation rates as LKF pixels, (ii) it separates the LKFs in a global binary map into minimal LKF-segments, and (iii) it re-connects multiple minimal segments into individual LKFs based on a probability that is determined by their distance, their orientation relative to each other, and their difference in deformation rates.

The tracking algorithm combines the detected LKFs of two subsequent time records with the drift information between the two records to track individual LKFs over time. First, the algorithm advects the LKFs from the first time record according to the drift information to obtain a first-guess positions for the LKFs. Then, tracked LKFs in the second record are identified by the degree of overlap between the advected LKFs and the detected LKFs of the second time record.

### 2.2 RGPS LKF-dataset

The deformation data of the RADARSAT Geophysical Processor System (RGPS, Kwok, 1998) was processed by the LKF detection and tracking algorithms (Hutter et al., 2019) to produce a comprehensive data set (https://doi.org/10.1594/PANGAEA. 898114, Hutter et al., 2019). The data set contains LKFs in the winter months (November to May) from 1996 to 2008 and covers large parts of the Amerasian Basin in the Arctic Ocean. In total the data set contains 165 000 detected LKFs and 36 000 tracked LKFs.

### 2.3 Model simulations

#### 2.3.1 Model configurations

Both simulations in this paper are based on a regional Arctic configuration (Nguyen et al., 2012) of the Massachusetts Institute of Technology general circulation model (MITgcm, Marshall et al., 1997; MITgcm Group, 2017), but with a refined horizontal grid spacing of 2 km. The number of vertical layers is reduced to 16 with the first five layers covering the uppermost 120 m to decrease computational cost associated with the ocean model component as we are only interested in sea-ice processes. The Refined Topography data set 2 (RTopo-2) (Schaffer and Timmermann, 2016) is used as bathymetry for the entire model domain. The lateral boundary conditions are taken from the globally optimized ECCO-2 simulations (Menemenlis et al., 2008). We use the 3-hourly Japanese 55-year Reanalysis (JRA-55, Kobayashi et al., 2015) with a spatial resolution of $0.5625°$ for surface

boundary conditions. In the baseline simulation, the ocean temperature and salinity are initialized on January 1st, 1992 from the World Ocean Atlas 2005 (Locarnini et al., 2006; Antonov et al., 2006). The initial conditions for sea-ice are taken from the Polar Science Center (Zhang et al., 2003). Ocean and sea ice parameterizations and parameters are directly taken from Nguyen et al. (2011) with the ice strength $P^\star = 2.264 \cdot 10^4 \, \text{Nm}^{-2}$. The baseline simulation uses the classical discrimination of two ice

classes: thin and thick ice (Hibler, 1979). The momentum equations are solved by an iterative method and Line Successive Relaxation (LSR) of the linearized equations following Zhang and Hibler (1997). In each time step ($\Delta t = 120 \, \text{s}$), 10 non-linear steps are made and the linear problem is iterated until an accuracy of $10^{-5}$ is reached, or 500 iterations are performed. The baseline simulation is run from January 1st, 1992 to December 31st, 2012. The analysis is based on daily averages of sea ice drift, ice thickness and concentration.

On October 17th, 1995 the simulation with an ice thickness distribution (Thorndike et al., 1975) with 5 thickness categories separated by boundaries at 0.0 m, 0.64 m, 1.39 m, 2.47 m, and 4.57 m is started. In doing so, the initial sea-ice thickness and concentration of all thickness categories need to be determined from the 2-category simulation for each grid-cell. Most commonly this conversion is done by assigning all ice in one grid cell to the category with the same ice thickness. Then, some years of spin-up time are used to redistribute the ice into different categories (Ungermann and Losch, 2018). Due to the

high-resolution in our simulation a multiyear spin-up is not affordable. Therefore, we use the fact that observed ITDs follow log-normal functions (Wadhams, 1992; Haas, 2010) and describe the ITD of each grid-cell by a log-normal distribution with a mode of $2/3$ of the mean thickness. The mean thickness and concentration over all categories remain unchanged. With this initialization of the ITD-simulations, the spin-up of the ice thickness distribution is reduced to one year. We use the ice strength formulation of Rothrock (1975) and the smooth partition and redistribution functions of Lipscomb et al. (2007). The simulation

with ITD is integrated from October 17th, 1995 to December 31st, 2012. In the following we refer to this simulation as "ITD" and to the baseline simulation as "noITD". Both models provide data in the RGPS period of 1996 to 2008.

Both model configurations are not tuned to reproduce the observed ice distributions due to limited computational resources. Instead, we carry over ocean and sea ice parameters from optimized coarse resolution configurations (Nguyen et al., 2011; Ungermann and Losch, 2018, for ITD specific parameters). The resulting simulations overestimate the seasonal amplitude of

sea ice volume and extent, but their trends are reasonable (not shown). The resulting ocean circulation has not been evaluated in detail, but the wind-driven surface circulation is plausible with strong mesoscale activity, the surface temperature and large scale sea ice distribution follow the prescribed surface forcing as expected. The main role of the ocean model is to provide dynamic bottom boundary conditions to the sea ice model.

### 2.3.2 Sampling and LKF extraction

The RGPS data set is originally provided as a Lagrangian data set that consists of trajectories of points that were followed throughout a winter season in consecutive SAR imagery. For each time record in this data set, vertices are constructed for four neighboring trajectories to approximate the deformation rates from the drift of a vertex by using line integrals. This results in Lagrangian deformation rates. Then, the Lagrangian data are interpolated onto a regular Eulerian grid that is the basis of the RGPS LKF data set.

This Lagrangian nature of the RGPS data set confounds the comparison to Eulerian model output. Different approaches have been used to overcome this issue: (1) generate Lagrangian trajectories by on- or offline advection of artificial buoys in the model simulation that are initialized at the initial position of RGPS trajectories (Rampal et al., 2016; Hutter et al., 2018), (2) interpolate simulated drift speeds to the position of the RGPS vertices (Spreen et al., 2016), or (3) just compare directly Eulerian gridded deformation rates with RGPS (Wang et al., 2016; Bouchat and Tremblay, 2017). For an accurate magnitude of the deformation rates and in particular the temporal scaling, only the most sophisticated option (1) can be used as it takes the advection of ice into account and addresses the effect of distorted vertices on the computation of the deformation rates (Lindsay and Stern, 2003) consistently for model and RGPS.

The LKF detection algorithm used here does not depend directly on the magnitude of deformation rate itself but on the local variations of the deformation rates (Hutter et al., 2019). Therefore, the detection algorithm can be applied directly to the deformation rates on the output grid assuming it is a regular grid. Thus we avoid complicated sampling strategies that involve expensive post-processing of model output. The advection of ice is taken into account by the tracking algorithm.

Deformation rates are computed from daily mean velocity output following the spatial discretization of strain-rates as formulated in the model code (Losch et al., 2010). We reduce the spatial resolution of the input fields of the detection algorithm by a factor of 3 to 6.75 km by taking only every third pixel into account to reduce computational costs. As deformation features in the simulations show a width of $\sim 5$ pixels this can be done without missing features in the detection. We detect features every three days to agree with the temporal resolution of RGPS. The parameters used in the detection algorithm are the same as in Hutter et al. (2019, their Tab. 1), where all parameters marked with [b] are scaled to the reduced model resolution by multiplying with a factor of $12.5\,\mathrm{km}/6.75\,\mathrm{km} = 1.85$ to account for the resolution difference between the simulations and the RGPS data set. The detected features are tracked with the tracking algorithm using the parameters of Hutter et al. (2019, their Tab. 2). The drift required for the tracking of LKFs is obtained by the integration of the mean daily velocities over a three day period. The LKF data-sets of both simulations can be found in Hutter (2019b, c).

## 2.4 Spatio-temporal scaling analysis

Sea-ice deformation is known to depend on spatial and temporal scales following a power-law (Weiss, 2013; Weiss and Dansereau, 2017, for spaced-time coupled form),

$$|\dot{\epsilon}(T,L)| \sim L^{\beta(T)}, \tag{1}$$

$$|\dot{\epsilon}(T,L)| \sim T^{\alpha(L)}, \tag{2}$$

where $|\dot{\epsilon}(T,L)|$ is the mean deformation rate for the temporal scale $T$ and the spatial scale $L$. These scaling properties have been used to compare the self-similarity of sea-ice deformation in satellite observations and various model simulations (Rampal et al., 2016; Spreen et al., 2016; Hutter et al., 2018; Bouchat and Tremblay, 2017). Higher moments of the deformation rate

also follow a power-law scaling (Marsan et al., 2004; Weiss and Dansereau, 2017),

$$|\dot{\epsilon}(T,L)^q| \sim L^{\beta(q)}, \tag{3}$$

$$|\dot{\epsilon}(T,L)^q| \sim T^{\alpha(q)}, \tag{4}$$

with $q$ being the order of the moment. Here, the scaling exponents vary with the moment order and follow quadratic struc-
ture functions $\beta(q) = aq^2 + bq$ and $\alpha(q) = cq^2 + dq$ to show the multi-fractal intermittency and localization, that is, larger
deformation events are more localized and intermittent that low deformation rates (Rampal et al., 2019).

The spatio-temporal scaling analysis performed in this paper is based on Lagrangian drift data as suggested in Section 2.3.2.
To transfer the RGPS sampling to the model output, we convert the regular gridded velocity output of the model to Lagrangian
drift data by integrating trajectories from daily averaged velocity output of the model. Virtual buoys are initialized on the RGPS
grid on November 1st of each year (1996-2007). The virtual buoys are advected with the modeled ice drift until mid-May of
the following year and their positions are recorded every day. Note that the integrated trajectories agree with RGPS drift data in
their initial position but not in their entire path. We use the initial position of the trajectories on the regular RGPS grid to define
rectangular cells of 4 buoys to compute deformation rates. These cells are followed over the entire winter. The deformation
rates at the finest (initial) scales of $L_0 = 10\,\mathrm{km}$ and $T_0 = 3\,\mathrm{days}$ are determined from the drift of the vertices using line integrals
(Lindsay and Stern, 2003). In this computation, cells are removed that change their size by a factor of two or more.

The deformation rates for larger spatial and temporal scales are obtained by averaging the original deformation rates — a
method referred to as coarse-graining (Marsan et al., 2004). First, we average $n^*$ deformation rates in the temporal domain
to obtain the deformation rates for $\dot{\epsilon}(T,L_0)$. The actual temporal scale of this average is determined by the number of valid
deformation rates at initial scale $n \leq n^*$ with $T = nT_0 \leq n^*T_0$. Averages for which $n \leq n^*/2$, that is, less than half of the
deformation rates in the interval, are available, are removed. Next, the deformation rates are averaged in space. In doing so,
we average $m^* \times m^*$ temporal averages of deformation rates to $\dot{\epsilon}(T,L)$. The spatial scale $L$ of the average is determined by
the square-root of the area that is covered by valid deformation rates where $L \leq m^*L_0$. All cells that are less than half filled
($L < L_0/2$) are removed.

## 2.5 Irregular temporal sampling of RGPS

The RGPS drift data sets consists of a set of points that are followed in time in consecutive images of RADARSAT. Therefore,
the temporal sampling of position updates depends on when the satellite passes over a particular area. The RADARSAT repeat
cycle is 3 days during which it covers the Arctic Ocean. The repeat cycle is also the general temporal resolution of RGPS.

Each time an image is available in a region (smaller than the entire Arctic) with RGPS drifters, the drifter positions are
updated. The time stamp of the record of all RGPS drifters within one image is the same, but the time stamp of the next
(subsequent) image is slightly delayed by the time it takes the satellite to fly to this neighboring region. Note that in this way
the positions of drifters that are on both images are updated twice within a time period much shorter than 3 days. The time
difference within one overfly (order of minutes) is small compared to the time difference between two different overflies that
cover the same region (order of 3 days). This irregular time sampling complicates the computation of deformation rates using

line integral approximations on polygons constructed of RGPS drifters, because all vertices (RGPS drifters) contained in the polygon need to have position records at the same time.

Comparing the deformation rates of RGPS data to model output leads to a second problem: we use trajectories initialized at RGPS positions and advected with daily mean model velocities as described in Sec. 2.4. The positions of these trajectories are saved in 3-day intervals, so that all deformation rates computed from model data cover a 3-day period and start and end at the same time. The deformation rates computed from RGPS data, however, have varying start and beginning time and varying time intervals.

Hence, the RGPS deformation rates need to be converted to regular 3-day intervals to set-up a common framework for a comparison. Rampal et al. (2019) used simple nearest neighbor interpolation to do so and glosses over temporal details. For a more accurate conversion, we take the following processing steps for each RGPS stream (all points that are covered by two consecutive overflies of the satellite, for details see Kwok and Cunningham, 2014): (1) we form rectangles from all RGPS drifters using their initial position. (2) For each rectangle, we check the time records of all vertices for time records that are shared by all four vertices with a tolerance of $\pm 3$ h. (3) From these common time records we compute velocity gradients if the time between two common records is larger than 1.75 days and smaller than 7 days. (4) The computed velocity gradients are then averaged in the fixed 3-day intervals weighted by the time they overlap with the fixed 3-day interval. The set of 3-day grided deformation rates for each stream is merged into a composite for the Arctic. In a region where two streams overlap, we choose the data of the stream that has a larger temporal coverage and remove the data of the other stream.

## 3 Scaling in sea-ice deformation

The mean deformation rates in the RGPS data set and both simulations decrease with the increasing spatial scale (Fig. 1a). These decreases follows a power-law (Eq. 3) showing that the deformation is strongly localized. The ITD simulation shows higher deformation rates than the RGPS data across all spatial scales, whereas the noITD simulation underestimates deformation rates. The spatial scaling exponents of the ITD simulation agree very well with RGPS observations. The noITD simulation shows a slightly weaker localization of deformation rates than the RGPS data. The scaling exponent increases with the moment order following a quadratic structure function for all three data-sets (Fig 1b). This shows that strong deformation events are more strongly localized than weak deformation events, which indicates multi-fractal spatial scaling of deformation rates. The structure functions have curvatures of $c = 0.14$ (RGPS) and $c = 0.15$ (both model simulations) consistent with previously published results ($c = 0.13 - 0.14$; Marsan et al., 2004; Rampal et al., 2016). The remarkable agreement of the curvature in RGPS data and model simulations implies that not only do the model simulations reproduce the spatial heterogeneity of deformation rates, but also the stronger localization of extreme deformation events.

The temporal scaling analysis (Fig. 2) shows that the sea-ice deformation is governed by the multi-fractal temporal scaling: (1) the moments of sea-ice deformation decrease with increasing temporal scale following the power-law of Eq. (4). (2) The temporal scaling exponents vary quadratically with the order of the moments. Again, the structure functions of the model simulations resemble the structure function found for RGPS data and the obtained curvatures $c = 0.13$ (ITD) and $c = 0.11$

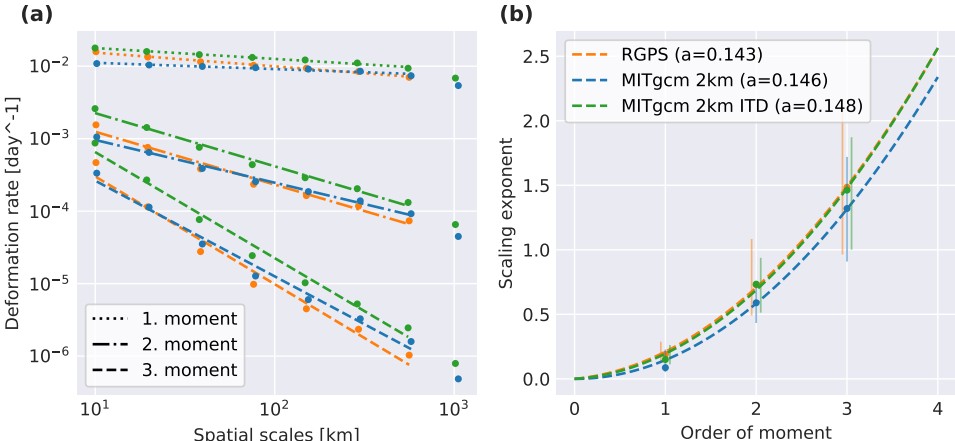

**Figure 1. (a)** The first three moments of sea-ice deformation rate as a function of the spatial scale for RGPS and both model simulations. **(b)** The spatial scaling exponents as a function of the moment order. A quadratic structure function $\beta(q) = aq^2 + bq$ is fitted (dashed lines). The curvature of the fit is given in the legend. The error bounds of the scaling exponents are determined by the minimum and maximum slope between successive points of the power-law fit.

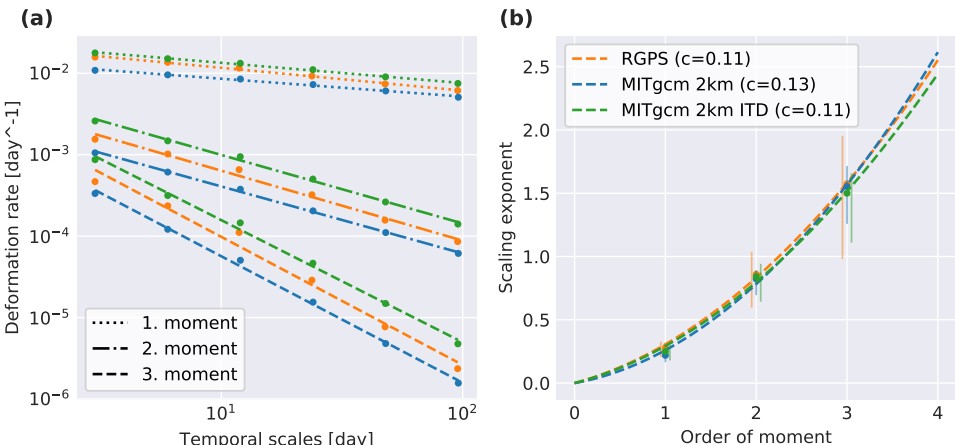

**Figure 2. (a)** The first three moments of sea-ice deformation rate as a function of the temporal scale for RGPS and both model simulations. **(b)** The temporal scaling exponents as a function of the moment order. A quadratic structure function $\alpha(q) = cq^2 + dq$ is fitted (dashed lines). The curvature of the fit is given in the legend. The error bounds of the scaling exponents are determined by the minimum and maximum slope between successive points of the power-law fit.

(RGPS, noITD) agree with previous studies ($c = 0.12$, Weiss and Dansereau, 2017). Again, a positive curvature of the structure function implies that the high deformation events are more strongly localized in time than small deformation events.

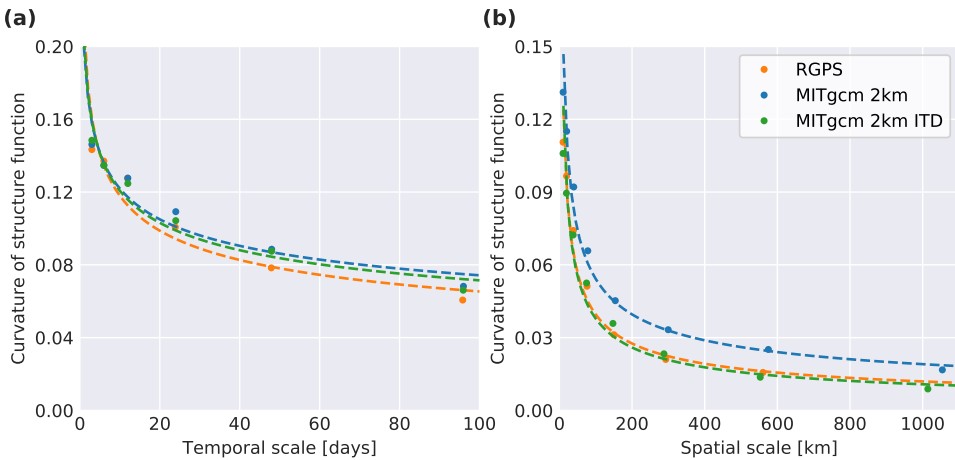

**Figure 3.** Spatio-temporal coupling of multi-fractal scaling: (a) The curvature of the structure function of the spatial scaling exponent as a function of temporal scale for RGPS data and both model simulations. (b) The curvature of the structure function of the temporal scaling exponent as a function of spatial scale. The dashed lines provide power-law fits to the decay of the curvature.

The multi-fractal scaling properties of sea-ice deformation imply a space-time coupling for both the RGPS data and model simulations, that is, the degree of localization changes with temporal scale and the degree of intermittency changes with spatial scale (Fig. 3). We find that for model simulations the curvature of the structure function of the spatial scaling exponent decreases with increasing temporal scale at a rate similar to the RGPS data (Fig. 3a). The curvature of the structure function of the temporal scaling exponent also decreases with increasing spatial scale (Fig. 3b). The curvature of the structure function of the temporal and spatial scaling exponent follows a power-law (Fig. 3a and b) as suggested by Rampal et al. (2019).

Comparing the spatio-temporal scaling of both model simulations, we find that deformation rates as well as spatial and temporal scaling exponents for the first moment of scaling deformation are higher in the ITD simulation compared to the noITD simulation. This supports the hypothesis of Hutter et al. (2018) that the ice strength formulation in the ITD run intensifies the feedback cycle of ice strength and deformation. This intensification in the ITD simulation is caused by the ice strength being more dependent on the concentration of the thinnest ice class. In the case of divergence, newly formed ice in open water fills this thinnest class and reduces the ice strength. Due to the reduced ice strength, deformation increases yielding to a stronger localization of deformation in space and time and thereby higher scaling exponents.

In summary, the spatio-temporal scaling analysis shows that both model simulations reproduce the observed multi-fractal heterogeneity and intermittency of sea-ice deformation (Marsan et al., 2004; Rampal et al., 2008; Weiss and Dansereau, 2017; Oikkonen et al., 2017) equally well as more sophisticated models that were specifically designed with these characteristics in mind (Girard et al., 2011).

## 4   LKF statistics

We split the analysis of the detected and tracked LKFs into three different parts: (1) the Pan-Arctic distribution of LKFs that describes the overall number of LKFs, the density and orientation of LKFs, (2) spatial properties of LKFs, that is, the length and curvature of LKFs as well as the angle at which they intersect, and (3) the temporal evolution of LKFs as described by their persistence and their growth rates. Each metric is presented and discussed in a separate subsection for RGPS data and both model simulations. The overall quality of LKFs in the simulations and the link between LKF statistics and scaling analysis is discussed in the separate Section 5.

This comparison includes some metrics that are sensitive to the coverage of the LKF data set. While the coverage of the model is Pan-Arctic and constant in time, the coverage of the RGPS data varies with time. We mask the LKFs of the model simulations with the RGPS coverage of the corresponding record. As some LKFs are removed in part or entirely, this filtering affects also the tracking of the features. The tracking algorithm is, therefore, run once again on the filtered features. We label the masked version of model LKFs in the legend of the corresponding plots.

### 4.1   Pan-Arctic distribution of LKFs

In the following section we test whether the model simulations reproduce (1) the number of features for different years and seasons, (2) the regional distribution of deformation features, and (3) the mean orientation compared to the RGPS LKF data-set.

#### 4.1.1   Number of LKFs

The first and most obvious metric for testing whether a model simulates LKFs in agreement with observations is the number of features detected in model simulations and observations. This metric was used for the optimization of solver parameters (Koldunov et al., 2019) and provides some valuable first insights. The number of detected features in RGPS data and both model simulations are given in Fig. 4. We, here, use the version of the LKF data set for both model simulations that has been filtered by the RGPS coverage and normalize the number of LKFs by the number of RGPS observations to account for the varying RGPS coverage.

The RGPS LKF data-set shows little variation in the number of deformation features in the entire observing period with no clear trend (from 0.015 to 0.0125 LKFs per RGPS observation). There is no significant seasonal variability in feature numbers for the RGPS data (Fig. 4b). The cumulative length of all LKFs, defined as the sum of all LKF lengths in one record, shows seasonal variations (Fig. 4d). The cumulative length decreases as the ice advances and reaches a minimum in mid March implying that in this season the area is smaller where atmosphere-ice-ocean interaction processes take place. This difference in the seasonal cycle shows that in the freezing season (November to January) LKFs tend to be larger than in the remaining part of the year.

The number of detected LKFs in the ITD simulation agrees on average very well with RGPS except for the last two winters (Fig. 4a). The interanual variability in LKF numbers, however, is larger by a factor of 2 compared to the RGPS data (shaded standard deviation in Fig. 4b). The number of features is higher than for the RGPS data in the early freezing period in November

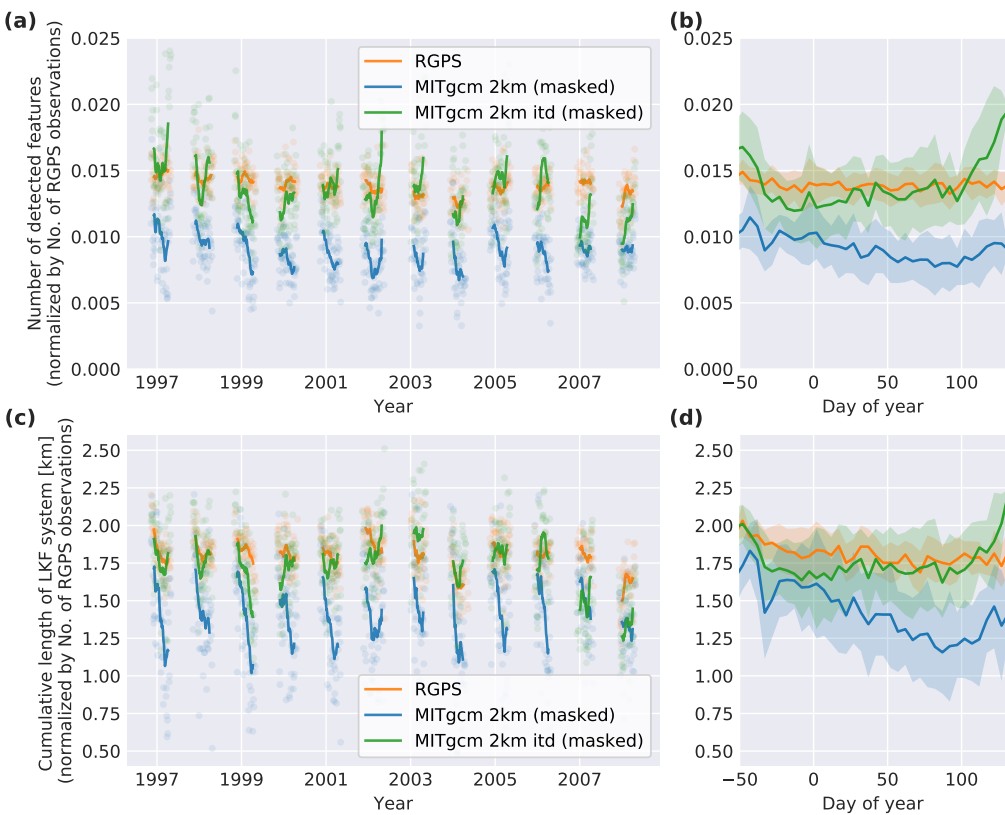

**Figure 4.** **(a)** The number of LKFs detected for entire observing period (1996 to 2008). The lines are running means with a window size of 17 days of the individual daily numbers represented by the light-colored dots. **(b)** Seasonality of the number of LKFs. The lines are running means with a window size of 5 days. The shaded areas show the standard deviation of the individual years. **(c)** Same as **(a)** but showing the cumulative length of the system of all LKFs, defined as the sum of all LKF lengths in one record. **(d)** The seasonal cycle of the cumulative length of the LKF system. For the analysis presented in this Figure the LKF data sets of both simulations are filtered for RGPS coverage.

and after mid April. Starting in late spring, the ice cover seems to be too weak in the simulation so that there are more LKFs and a generally too large LKF network.

In the noITD-simulation, there are $\sim 30\%$ fewer detected features than for the RGPS data set and the cumulative LKF length is shorter by $\sim 20\%$. This simulation reproduces the weak seasonal cycle of the numbers of LKFs in the RGPS data, but numbers are generally too low and then they slightly increase in April (Fig. 4b and d). For the cumulative LKF length, however, the magnitude of seasonal cycle is twice as strong for the model than for the RGPS data (Fig. 4c), which shows that the average

LKF length in the simulation is more variable. The noITD simulations shows a higher average LKF length compared to RGPS data before March and a lower average LKF length afterwards.

### 4.1.2 LKF density

The number of LKFs only provides insight about the temporal development of LKFs, but the LKF densities (i.e. their relative
frequency of occurrence) show how the simulated LKFs vary in space compared to the RGPS data. In the RGPS data (see also Hutter et al., 2019), we find the highest densities along the shorelines of islands such as the New Siberian Islands and Wrangel Island (Fig. 5a; for geographical reference see Fig. 5b). The highest densities within the ice pack are found in the Beaufort Sea. Low densities stand out in the fast-ice region in the East Siberian Sea with a sharp fast-ice edge. Similar distributions were found for lead densities derived from MODIS thermal-infrared imagery (Willmes and Heinemann, 2016) and CryoSat-2 data
(Wernecke and Kaleschke, 2015).

In both simulations, the regions of high LKF densities are similar, but the ITD simulations has generally higher densities. Here, LKFs concentrate along small islands and coastlines as in the RGPS data. Besides Wrangel Island and the New Siberian Islands, Severmaya Zemlya and Franz Josef Land are preferred starting points of LKFs. The highest densities along the coastlines are found at Barrow (Alaska) and at North-Eastern tip of Greenland, consistent with remote sensing data es-
timates (Willmes and Heinemann, 2016). These high densities along the coast of Alaska are not resolved in the RGPS data set, because the detection algorithm cannot identify LKFs that are located at the edge of the RGPS coverage. The general overestimation of coastal deformation in the model simulations combined with an underestimation in the pack-ice compared to RGPS data suggests stress propagation to the coast in the model due to a lack of inhomogeneities in the pack-ice that serve as seeding points for failure.

In both simulation we observe distinct fast-ice regions with low LKF densities only in the Eastern Laptev Sea, but not in the East-Siberian Sea. Further, the simulated LKFs do not accumulate at the Hanna Shoal as the RGPS LKF densities. In this shallow region, keels of pressure ridge frequently ground which initiates the formation of leads (Mahoney et al., 2012). Both the missing landfast ice in the East-Siberian Sea and the missing effect of grounded ice may be improved by implementing a grounding scheme (Lemieux et al., 2015).

For the RGPS data, a band of enhanced LKF activity connecting Hanna Shoal and Wrangel Island (Fig. 5) is consistent with results based on Advanced Very High Resolution Radiometer (AVHRR) (Mahoney et al., 2012). In the RGPS data, we observe lower LKF densities north of this band. Neither the band nor the region of low LKF densities can be found in either simulations. We speculate that in the model there are no stress locators in the form of grounded ice, which leads to a spatially broader distribution of failure and LKFs.

The elevated LKF densities in the Beaufort Sea in the RGPS data are another prominent feature within the pack ice. These have been attributed to the shear induced by the Beaufort Gyre circulation (Willmes and Heinemann, 2016). We do not observe the increased probability in LKF formation in either simulations, which may suggest that the Beaufort Gyre circulation is too weak, there are too few mesoscale eddies (Zhao et al., 2014), or that the ice-ocean drag parameterization that does not take into account keels and sails in deformed multi-year ice is too simple (Tsamados et al., 2014; Castellani et al., 2018).

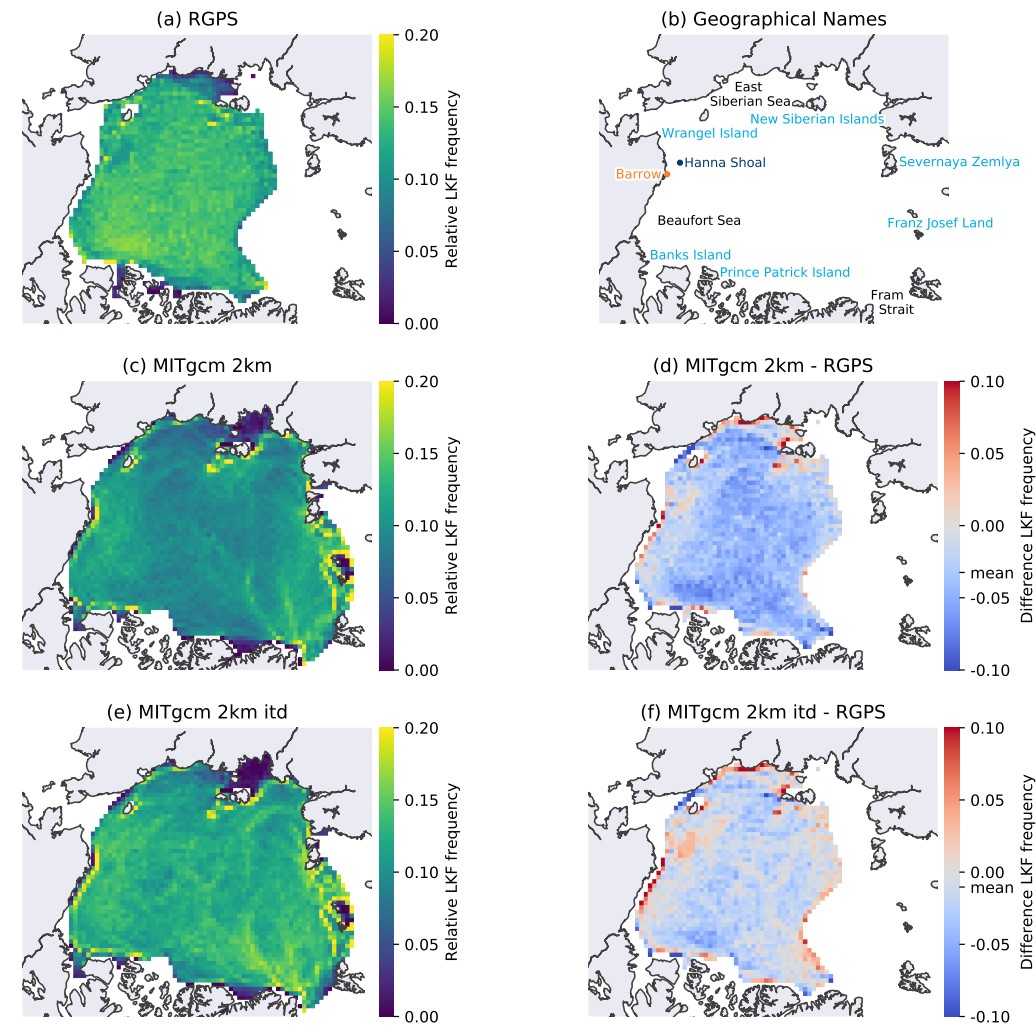

**Figure 5. (a,c,e)** The density of LKFs in the RGPS data set and the two model simulations for the winters between 1996 and 2008 computed in $50 \times 50$ km boxes over the Arctic Ocean. The absolute frequency of LKF pixels in a box is normalized by the total number of pixels with deformation in the data set. Only boxes with more than 500 deformation pixels in space and time are shown. **(d,f)** The difference in density between both model simulations and RGPS. **(b)** The geographical locations that are referenced in the text.

### 4.1.3 LKF orientation

A preferred LKF orientation indicates frequently occurring forcing conditions. We compute the orientation of LKFs clustered in $200\,\text{km} \times 200\,\text{km}$ boxes in the Arctic. Within each box, the orientation of the part of each LKF that overlaps with the box is determined. From these orientations we compute the mean orientation (Fig. 6a-c) following Bröhan and Kaleschke (2014). In addition, we determine the modes of the orientation distribution within each box (Fig. 6d-f), as the mean orientation is misleading for multi-modal distributions. We find the modes of multi-modal distributions of LKF orientations by determining local maxima in the Probablitiy Density Function (PDF). First, we estimate the PDF of the distribution using a Kernel Density Estimation (KDE) with von-Mises kernels that can be seen as the analogue of the Gaussian distribution on a circular domain (Borradaile, 2003). The modes of the distribution are given by the local maxima of the PDF estimated by the KDE. To test whether the obtained orientation distributions are significantly different from a random distribution, we draw 10,000 random orientation samples of the same size and perform a $\chi^2$-test (Bröhan and Kaleschke, 2014). We mark points where the mean probability of the $\chi^2$-tests is less than $1\,\%$ as statistically significant. We take into account only LKFs with an average total deformation rate of $0.5\,\text{day}^{-1}$ to agree with visually identified LKFs (Kwok, 2001).

Both the mean orientation and the first mode of orientation are generally parallel to the coastline (Fig. 6a and d). The LKFs orientated in this direction are most likely flaw leads between the fast ice along the coast and the mobile pack ice. The model simulations reproduce this behavior along all coastlines. The parallel orientation to coastlines is consistent with lead orientations reported earlier (Miles and Barry, 1998; Bröhan and Kaleschke, 2014).

In the east Siberian Sea the mean LKF orientation is from east to west in the RGPS data as well as in both model simulations. This is also the orientation of the local fast-ice edge. The mean orientation shifts towards the north-south direction in the Beaufort Sea, again for the observations and both simulations. This general pattern of LKF orientation is consistent with the orientations derived from visually detected leads in thermal- and visible-band imagery of the years 1979–1985 (Miles and Barry, 1998). Interestingly, there are substantial differences to more recent results with predominantly east-west orientations in the Beaufort Sea (Bröhan and Kaleschke, 2014). These differences may appear because Bröhan and Kaleschke (2014) used data from a different time period (years 2002–2011) or because they used a more sophisticated statistical method (Hough transform). We note that although RGPS data and simulations agree in the mean LKF orientation in these regions, the model shows a spread in the modal values, which points towards too large LKF intersection angles that lead to two peaks in the distribution of orientations.

In the Central Arctic, the mean orientation of LKFs in the RGPS data suggests a circular deformation pattern that originates in North-East Greenland, circles the North Pole and heads towards Severnaya Zemlya. The pattern is reminiscent of a basin-scale ice arch formed by the main sea ice export pathway through the Fram Strait. In the modal representation of LKF orientations, this arch is barely visible in the model simulations. In the Fram Strait, we find modes parallel and perpendicular to the outflow direction, indicating the build-up of flow blockages similar to ice arches together with shearing zones between the exported sea ice and the fast-ice along the coast of Greenland.

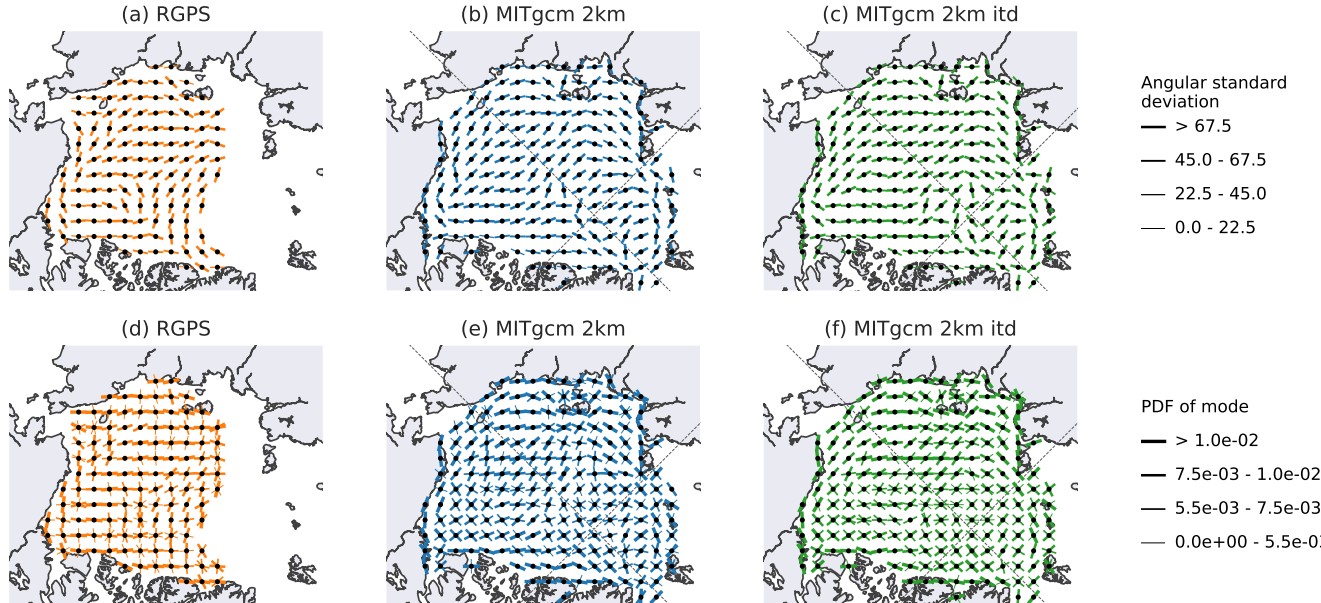

**Figure 6. (a,b,c)** Mean orientation of LKFs for RGPS and two model simulations for the winters between 1996 and 2008. We follow the definition of mean and standard deviations for circular data from Bröhan and Kaleschke (2014) for lead orientations. The line width indicates the standard deviation of the distibrution within the cell. Orientation distributions that are statistically significantly different from a random distribution are marked by a black dot. Only cells that contain more than 500 LKFs are shown. **(d,e,f)** Modal orientation of LKFs for RGPS and two model simulations. The three largest modes of the distribution of orientation are plotted for cells with including more than 500 LKFs. The probability (PDF) of each mode is shown by the line length, where a PDF value of $5.5 \cdot 10^{-3}$ corresponds to the mean PDF value of a random distribution. In **(b,c,e,f)** the $0°$, $90°$, $180°$, and $-90°$-meridian are shown as dashed grey lines. The numerical grid lines of both simulations are parallel to these meridians.

## 4.2 Spatial LKF properties

### 4.2.1 LKF length

LKFs in the Arctic have length scales from a few meters up to the basin-scale (1000 km). Given that automated lead detection is challenging and hand-picked lead data-sets have a limited sampling size, the first quantitative estimates of LKF length
5    have been published only very recently (Linow and Dierking, 2017; Hutter et al., 2019). From 10 RGPS records Linow and Dierking (2017) inferred an exponential distribution of LKF lengths. Length measurements of lead skeletons, again from a small sample-size, were also distributed exponentially (Van Dyne et al., 1998). The distribution of LKF lengths from the entire RGPS data-set contains more extreme values (Hutter et al., 2019) and is described by a stretched-exponential distribution: $p(x) = Cx^{\beta-1}e^{-\lambda x^{\beta}}$ with $C = \beta\lambda e^{\lambda x^{\beta}_{\min}}$ (Clauset et al., 2009). Stretched-exponential distributions belong to the family of

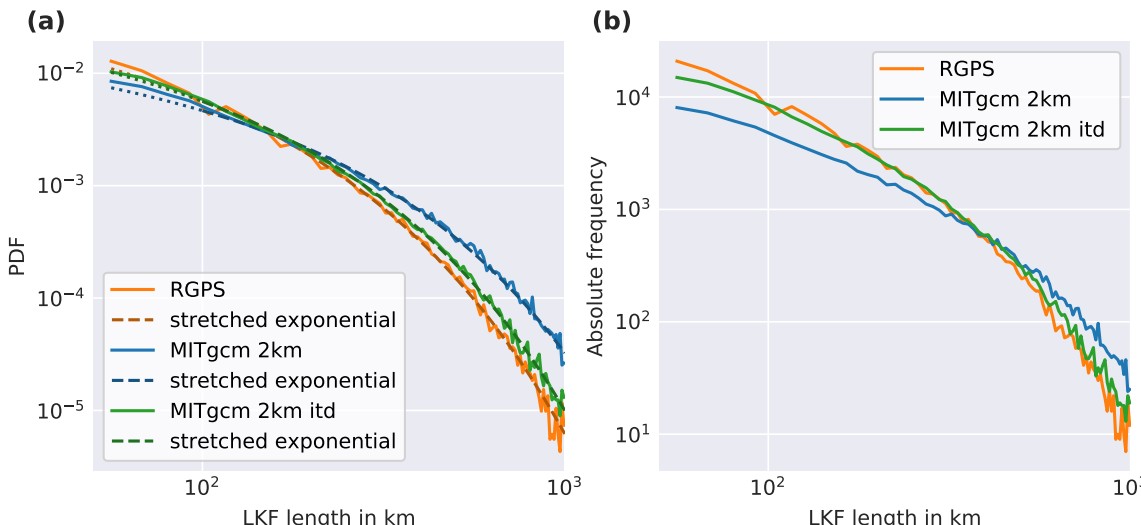

**Figure 7. (a)** PDF of LKF lengths for RGPS data and both simulations along with fits to stretched exponential distributions. **(b)** Absolute frequencies of LKF lengths in RGPS data and both simulations. For the analysis presented in this Figure the LKF data sets of both simulations are filtered for RGPS coverage.

heavy-tailed distribution, but in contrast to the power-law distribution have a natural upper limit scale (Laherrère and Sornette, 1998). Here, the upper limit scale is the finite size of the Arctic Ocean.

We determine the PDF of LKF lengths from RGPS data and both model simulations (Fig. 7a). The length is measured as the cumulative sum of the distance between pixels along the LKF. We fit a stretched exponential distribution to all data-sets using Maximum Likelihood Estimators and perform a goodness-of-fit test (Hutter et al., 2019). We find that all distributions are accurately described by stretched exponentials (RGPS: $\lambda = 1.69 \cdot 10^{-2}, \beta = 0.719$, noITD: $\lambda = 0.90 \cdot 10^{-2}, \beta = 0.761$, and ITD: $\lambda = 1.38 \cdot 10^{-2}, \beta = 0.741$). The PDF of the ITD simulation agrees remarkably well with the RGPS PDF; only for LKFs longer than 150 km, the probabilities are slightly higher. In the noITD simulation large LKFs ($> 300$ km) have a higher probability than in the RGPS data or in the ITD simulation. From the absolute frequencies of LKF lengths (Fig. 7b), it becomes clear that these high values in the (normalized) PDF are to a large extent a consequence of too few small LKFs. From the deficit in small-scale deformation features in the noITD simulation one may infer that there are too few inhomogeneities in the ice cover that can initiate failure. The ITD sub-grid model allows for more small-scale variations in the ice thickness distribution that are reflected in the ice strength and in turn lead to stronger localized deformation.

### 4.2.2 LKF curvature

As their name implies, Linear Kinematic Features are mostly linear with only little curvature (Kwok, 2001). Linow and Dierking (2017) introduced the dependence of the distance between both endpoints on the LKF length as a metric for the curvature

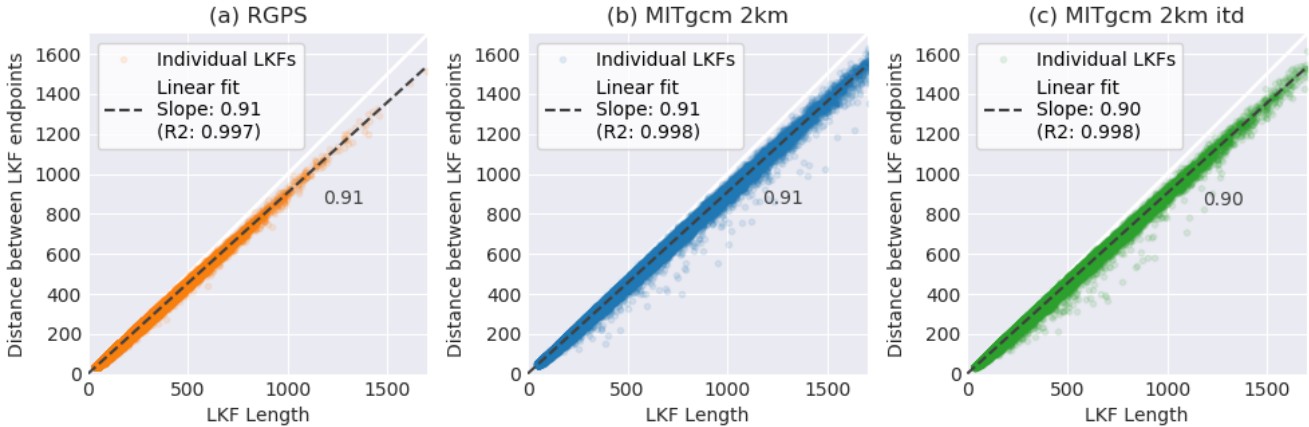

**Figure 8.** The distance between the endpoints of an LKF plotted over its length. We find that all observations fit a linear line. The slope of the fitted linear relationship is given next to the fitted line.

of an LKF. This ratio is 1 for perfectly linear features and 0 for circular ones. We apply this metric to all LKFs detected in the RGPS data and both simulations and plot the distance between LKF endpoints against the LKF length (Fig. 8). For perfectly linear LKFs, all points converge to a line with a slope of 1 (the diagonal of the plot). We find that all LKFs clearly fall on a line with slopes of 0.91 (RGPS and noITD) and 0.90 (ITD) for all length scales from up to 1700 km. Thus, the shape of the LKFs

does not change with spatial scale, which agrees very well with the self-similar properties of sea-ice shown for the size of ice floes (Stern et al., 2018) and deformation features (Weiss, 2003; Marsan et al., 2004; Wernecke and Kaleschke, 2015). Both simulations reproduce the curvature of the features itself and its scale invariance. The spread in curvature decreases at smaller LKF length ($< 200$ km) for RGPS and both simulations, because the LKF detection algorithm does a poor job of detecting short high-curvature LKFs. We, here, note that this fairly simple metric does not allow further inferences about the shape of

LKFs, but it has the advantage that it can be applied in a straightforward manner to a large variety of LKFs.

### 4.2.3   LKF intersection angles

The intersection angle of LKFs is strongly related to the material properties, more precisely to the yield curve of the rheology (Erlingsson, 1988; Wang, 2007), but only intersection angles of conjugate faults, where intersecting LKFs form instantaneously under the same forcing conditions, provide direct information about the yield curve. Therefore, we limit the analysis of inter-

section angles to pairs of LKFs that form in the same time record. We note that with this restriction the maximum time between the formation of both LKFs is determined by the temporal resolution of RGPS of 3 days. Therefore some LKF pairs may not

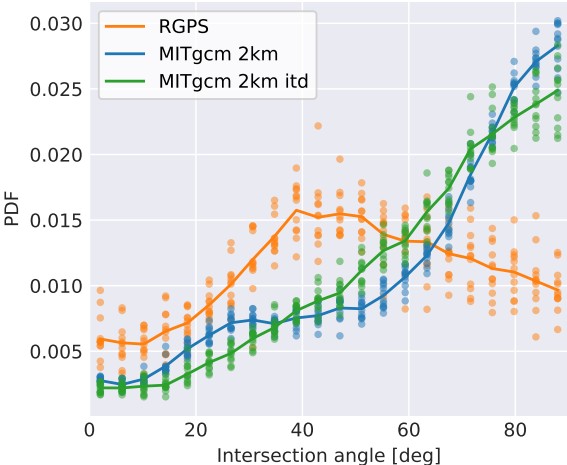

**Figure 9.** The PDF of intersection angles of LKFs in RGPS data and both simulations. The analysis is limited to pairs of LKFs that formed in the same time record and have a length larger 125 km. Individual years are plotted as colored points and the multi-year mean is given as a solid line.

have formed simultaneously. In addition, we require the length of both LKFs to be larger than 125 km to exclude the effect of a preferred direction along the pixels of the image.

The PDF of intersection angles for RGPS data peaks around $40°$–$50°$ (Fig. 9). This peak agrees with typical intersection angles of $30°$–$50°$ inferred from satellite imagery (Walter and Overland, 1993; Cunningham et al., 1994; Schulson, 2004; Wang, 2007) and laboratory measurements (Schulson et al., 2006). We find the lowest probabilities for angels smaller than $20°$. Angles larger than $50°$ occur more often than angles smaller than $40°$. The distributions of intersection angels in both model simulations are very different from the RGPS data and peak at $90°$, which is in agreement with idealized experiments using the VP rheology (Hutchings et al., 2005). Intersection angles smaller than $60°$ are less frequent in the model simulations than in the RGPS data. The differences between both simulations are small.

According to theoretical considerations, the intersection angle is determined by the slope of yield curve (Pritchard, 1988; Ukita and Moritz, 1995; Wang, 2007). As both simulations use the same elliptical yield curve with a normal flow-rule (Hibler, 1979) similar intersection angles of LKFs are expected. We attribute the small differences in Fig. 9 to sea ice fields with a different amount of LKFs. Ringeisen et al. (2019) derived for idealized compression experiments that it is impossible to obtain intersection angles smaller than $60°$ with an elliptical yield curve. This explains the deficit of small intersection angles in our simulations. The peak in the PDF near $90°$ suggests a dominant LKF alignment with the numerical grid. A close inspection, however, does not show this dominant alignment (Fig. 6 b,c,e,f). Therefore, we can assume that the grid orientation has only a small effect on the LKF orientation, but that the rheology itself causes the overestimation of the intersection angle.

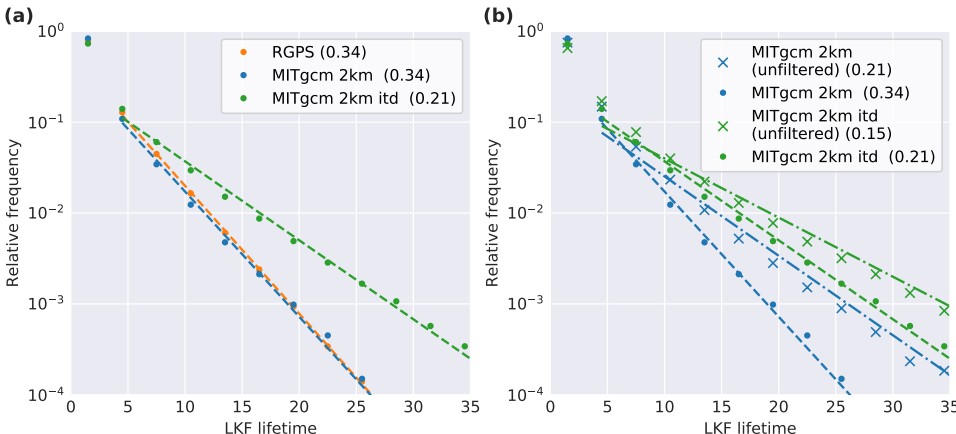

**Figure 10. (a)** The relative frequency of lifetime of LKFs in RGPS data and both simulations. The LKFs detected and tracked in the simulations are reduced to the RGPS coverage. The dashed line are fits to exponential tails. The rate of the exponential tail in day$^{-1}$ is given in the legend. **(b)** Lifetime of modeled LKFs reduced to the RGPS coverage (dots and dashed lines) and unfiltered (crosses and dash-dotted lines).

## 4.3 Temporal evolution of LKFs

The temporal evolution of LKFs has not been studied very much. In-situ field observations of individual leads breaking individual floes (e.g. Dempsey et al., 2012) suffer from space and time limitations. Qualitative evaluations of the persistence of lead patterns on the order of a month (Kwok, 2001) did not focus on individual leads to deduce these temporal characteristics
of LKFs and are necessarily inaccurate. Hutter et al. (2019) combined the large coverage of the RGPS data with a tracking algorithm to provide first qualitative estimates of LKF lifetimes. Here, we use the same method to study the persistence and the growth rates of LKFs.

### 4.3.1 LKF persistence

We determine the lifetime of an LKF by counting how many times we track a feature. The lifetime estimates are binned into
3-day intervals, that is, the temporal resolution of the deformation data. If an LKF can not be tracked, we assign it to the lowest lifetime class ($0 - 3$ days). Tracked LKFs are assigned to a lifetime class according to the number of tracks (one time tracked is assigned to $3 - 6$ days, two time tracked to $6 - 9$ days, etc.). We compute the lifetime of LKFs in RGPS data and in both simulations. For the simulations we provide two calculations each: one after reducing the simulation data to the RGPS coverage (Fig. 10a) and one for the full data sets (Fig. 10b). All lifetime distributions have an exponential tail. For the comparison with
LKF lifetimes in the RGPS data we use the reduced versions of the model simulations (Fig. 10a). The lifetime distribution of the noITD simulation and the RGPS data have the same rate of the exponent tail of $0.34\,\mathrm{day}^{-1}$. The ITD simulations overestimates the lifetime of LKFs with an exponential tail decaying with the rate of $0.21\,\mathrm{day}^{-1}$.

The long LKF lifetimes in the ITD simulation can be caused either by too homogeneous forcing fields or a too strong memory of past deformation, both of which favor continuous deformation of ice. Given the high temporal and spatial resolution of the atmospheric forcing data, too homogeneous forcing fields are unlikely to be the only cause. In the ITD simulation, deformation events imprint on the ice thickness distribution of a grid cell, which has an effect on the ice strength and hence leads to fast feedbacks on the deformation itself. In this sense, these changes in the ice thickness distribution can be regarded as a memory in the ice. Interestingly, this extra memory with the ice strength of Rothrock (1975) improves the agreement with the number of LKFs (Sec. 4.1.1) and the LKF density (Sec. 4.1.2) in the RGPS data set, but reduces the agreement with the RGPS data set in terms of LKF lifetimes. This suggests to improve the sea-ice model by decoupling the ice memory in the model from the ice thickness distribution.

The varying spatial coverage of the RGPS data introduces an unknown bias in LKF lifetimes. With our simulations we can estimate this bias by comparing the lifetimes of filtered LKFs to unfiltered LKFs (Fig. 10b). The difference for the noITD simulation, which agrees almost perfectly with RGPS data when masked by RGPS coverage (Fig. 10a), suggests that the amount of long lifetimes is reduced by the varying coverage with the rate of the exponential tail decreasing to $0.21\,\mathrm{day}^{-1}$, which corresponds to an increase in mean lifetime of $\sim 50\,\%$. For the ITD simulation the effect is similar.

### 4.3.2 LKF growth rates

Failure propagates quickly through the sea ice cover. This propagation can be modified or even stopped by changing forcing conditions. The growth rates of persistent LKFs provide information about these processes. We define the growth rate as the change in length of an LKF divided by the time between two records. In detail, we compute the area where both LKFs of a tracked pair overlap following the definition of overlap from Hutter et al. (2019) to determine how much of the change in length is attributed to growth and shrinking. This overlapping area is the part of the LKF that is seen in both time records. All parts of the LKF from the first time record that do not lie in the overlapping area are parts of the LKF that become inactive in the next time record. We associate the shrinking rate to these changes. Analogously, the LKF grows by the parts of the LKF in the second time record that lie outside of the overlapping area. These changes are associated with the (positive) growth rate. For completeness we also compute the growth rates of newly formed LKFs as their initial length divided by the temporal resolution.

All three growth rates follow an exponential distribution for the RGPS data and both model simulations (Fig. 11). Positive growth has the largest growth rates and the slowest decay of the exponential tail. The growth rate distributions of newly formed LKFs have the steepest exponential tails, but a higher probability of small growth rates ($< 50\,\mathrm{km/day}$). This implies that it is more likely for an existing LKF to grow longer than for a new one to form. From a physical point of view, this is plausible because an existing LKF is a weakness in the ice, where, with constant forcing, stress can accumulate which facilitates further deformation. Both model simulation contain this effect.

The shrinking rates of persistent LKFs are smaller than their growth rates and larger than the growth rate of newly formed LKFs. The physical interpretation is that the fracture of ice acts on much smaller time scales than the healing of the ice cover. Therefore, breaking the ice and opening a lead takes less time than closing the lead by refreezing or convergent ice motion.

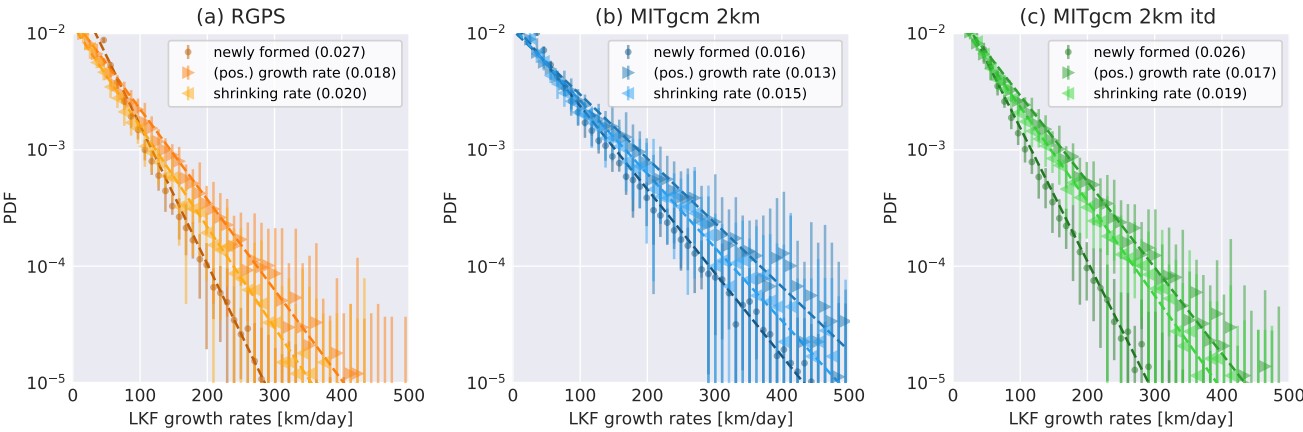

**Figure 11.** Mean (symbols) and standard deviation (vertical bars) of the growth rates of LKFs in RGPS data and both simulations. We differentiate between the growth rate, the shrinking rate, and the growth rate of newly formed LKFs. The dashed lines are fits to exponential tails, the rate parameter of which is given in the legend. For the analysis presented in this Figure the LKF data sets of both simulations are filtered for RGPS coverage.

Both mechanisms — higher growth rates of persistent LKFs and slower closing of LKFs — are present in RGPS data and both model simulations. However, only the ITD simulation reproduces the higher rate parameters of the exponential tails of RGPS observations. The tails for the noITD simulation decay more slowly for all three growth rate distributions. We attribute these differences to two effects: (1) the feedback between deformation and thickness that accelerates the formation of LKFs is

slower without the ITD due to the different strength formulation. Therefore, higher stress builds up before plastic deformation takes place, and consequently larger LKFs are formed. (2) The low LKF density in the noITD simulation leaves enough space for LKFs to grow larger. There are two plausible ways of stopping an LKF from growing: either the stress is too small for further growth, or the LKF intersects another LKF. Thus, higher LKF densities lead to lower growth rates.

     The low temporal resolution of 3 days cannot resolve true growing or shrinking rates of the LKFs, because the formation

or destruction of a deformation features may take place on much shorter time scales. Therefore, the growth rates computed here cannot be compared to fracture speeds of sea ice determined from acoustical measurements. These fracture speeds range from 100 to 1100 m/s (Stamoulis and Dyer, 2000) and can be regarded as an upper limit of the LKF growth rates. In-situ measurements of lead propagation times can also be used as an upper limit: Dempsey et al. (2012) recorded a time of 10 s to break a 80 m ice floe which corresponds to a fracture speed of $\sim 690$ km/day. A higher temporal sampling rate is needed to

directly compare our LKF growth rates with fracture speeds, which could be achieved by a higher output frequency for the model but is not possible for satellite observations.

## 5 Discussion

In this section we discuss how the new feature-based evaluation of LKF statistics is linked to the scaling analysis and what insights can be gained from LKF statistics for further model development.

Both simulations with and without ice thickness distribution (ITD) model agree remarkably well with satellite observations with respect to the representation of LKFs and spatio-temporal scaling analysis of sea ice deformation. We find that the simulated sea-ice deformation reproduces the multi-fractality in both spatial and temporal scaling, as well as the spatio-temporal coupling of multi-fractal characteristics, which remained a challenge even for simulations with the MEB-rheology (Rampal et al., 2019). Hence, our simulated sea ice deformation is characterized by strong heterogeneity and intermittency. Our present analysis is the most extensive scaling analysis so far for simulations with the VP-rheology and completes a set of previous scaling analyses (Bouchat and Tremblay, 2017; Spreen et al., 2016; Hutter et al., 2018) that show that previous findings about VP-simulations not being able to reproduce observed scaling properties at coarse grid resolution (Girard et al., 2009) cannot be generalized (especially not to high grid-resolution simulations). For mean sea-ice deformation (first order moment $q = 1$) the simulation with an ITD leads to scaling exponents closer to the ones retrieved from RGPS, that is, sea-ice deformation is more strongly localized in space and time compared to the simulation without an ITD. We attribute this result to the ice strength parameterization of Rothrock (1975) in the ITD run that favors a fast evolution of plastic deformation.

Consistent with the scaling analysis in Section 3, we find that the ITD simulation has a larger number of LKFs compared to the noITD simulation. In general, the LKFs in the ITD simulation agree better with RGPS data in terms of the LKF statistics. The LKF lifetime is the only exception. We note that both simulations reproduce all observed characteristics: heavy-tailed distribution of LKF lengths, exponential tails in the distribution of LKF lifetimes and growth rates, and scale invariance of the LKF curvature. The distribution of intersection angles, however, is not reproduced in either simulations.

By combining the scaling analysis and the feature-based evaluation, we test which inferences from the scaling properties can be made about the representation of LKFs. We discuss in the following how the results of both analyses are linked in our special case, but stress that one analysis cannot replace the other. Nevertheless, their combined use may provide new insights from previous scaling studies.

The largest difference in LKFs between the simulations is that the noITD simulation produces considerably fewer LKFs compared to the ITD simulation. In Sec. 4.2.1, we discussed how this overall underestimation influences LKF properties such as LKF length and growth rates. In addition, deformation fields that include fewer features of localized deformation obviously will be smoother in both space and time. We attribute the lower scaling exponents for the noITD simulation largely to the lower numbers of LKFs. The multi-fractal spatial scaling for both simulations is consistent with the heavy-tailed distribution of LKF lengths and the scale invariance in LKF shapes that suggest self-similar deformation patterns.

The difference in temporal scaling exponents also seems to be caused largely by the difference in LKF numbers because the noITD simulation reproduces the distribution of LKF lifetimes remarkably well (Fig. 10a) in spite of a lower temporal scaling exponent (Fig. 2). In contrast, the rate parameter of the LKF lifetime distribution in the ITD simulation is too low (Fig. 10) despite the better temporal scaling exponent. The absolute number of short-lived LKFs in the ITD simulation, however, is

higher and thereby closer to estimates from RGPS data just because of the higher number of LKFs. In this case, linking the results of the temporal scaling analysis to the representation of LKF lifetimes as attempted in Hutter et al. (2018) appears to be incorrect.

The agreement of the scaling analysis and the LKFs statistics between the model simulations and RGPS data may appear almost surprising given that the models have not been tuned at all for these diagnostics. We argue that this model performance is not determined by the large scale distribution of sea ice thickness and concentration, but the plastic model physics. For the plastic physics in VP-models to produce highly intermittent and heterogeneous LKF distributions, high resolution (Spreen et al., 2016; Hutter et al., 2018) and a sufficiently accurate solver (Koldunov et al., 2019) are necessary. As long as there is a quasi-closed ice cover, which is the case where and when the RGPS data are available, the plastic physics will produce localized deformation — even in idealized configurations (Hutter, 2015; Heorton et al., 2018) — and the associated statistics.

Both model simulations use the same grid and the same atmospheric forcing, which precludes direct inferences of resolution impact on the presented statistics. Here, we comment on expected impact based on previous studies. With increasing horizontal grid spacing, deformation features are more localized and more frequent (Spreen et al., 2016). Thus the number of LKFs, presented in Section 4.1.1, are likely to increase with model resolution along with a decrease in LKF length and growth rates as discussed in Section 4.2.1 and Sec. 4.3.2. In idealized experiments, it has been shown that higher spatial resolution of the atmospheric forcing also has the potential to increase the localisation of sea-ice deformation (Hutter, 2015). This suggests that also the number of LKFs increase. We speculate, however, that in our simulations this effect is saturated because we already use atmospheric forcing with fairly high resolution (JRA-55, $0.5625°$) that resolves most scales associated with the wind. To our knowledge, there is no study on the impact of temporal resolution on sea ice deformation. We hypothesize that an increased temporal resolution of the forcing will increase the short-term variability in the evolution LKFs, with direct impact on the LKF growth rates and presence of short-lived LKFs. However, please note that the order of this short-term variability is given by the temporal resolution (hours), which is much smaller than the shortest LKF lifetimes regarded in our study ($0 - 3$ days). We suggest to disentangle the effects of model resolution and wind forcing, but also ice strength parameterisation and solver parameters, on the formation of LKFs in a sensitivity study. For such a study, idealized experiments appear most suitable as they easily allow for higher number of simulations and to isolate effects.

With the LKF statistics, we identified some issues in the simulated deformation fields that could be addressed by specific parameterizations. A general theme in the discussion of LKF characteristics is the low LKF density in the simulations, which we attribute to too few inhomogeneities in the ice that can act as a starting point for fracture. With an ITD, the number and density of LKFs increase significantly. In the ITD simulation shear and divergence have a strong impact on the thin thickness classes which immediately feeds back into the ice strength facilitating further deformation. Therefore, inhomogeneities introduced by deformation in the thickness fields are much stronger compared to the standard VP simulation.

Simulated LKF densities in the pack ice away from the coast are too low compared to RGPS data and the distribution of LKF lifetimes is biased towards long-lived LKFs. The results imply that introducing inhomogeneities by using an ITD model may be one way of improving the model, but not necessarily the best one. The increased presence of long-lived LKFs in the ITD simulation suggests that one should reduce the strong feedback between ice thickness change and ice strength and instead

introduce a damage parameter. This damage parameter would act as the memory of past deformation and would also feed back into the ice strength, similar to the one used in EB/MEB models (Girard et al., 2011; Dansereau et al., 2016). Note that the local degree of anisotropy of the elastic-anisotropic plastic rheology (Tsamados et al., 2013) also represents a memory of past deformation. In doing so, a properly parameterized healing time could be tuned independently from parameters of the ITD formulation. We stress that a systematic parameter optimization (e.g. Massonnet et al., 2014; Ungermann et al., 2017; Sumata et al., 2019) is beyond the scope of this paper and also not possible due to limited computing resources.

The underrepresentation of fast-ice and LKFs that start from anchor points at shoals could be addressed with a grounding scheme (e.g. Lemieux et al., 2015). As the missing anchor points at shoals lead to an overestimation of LKF densities in the Chukchi Sea, this parameterization may also have an impact beyond improving the representation of fast-ice and LKFs at shoals.

Although the model reproduces most LKF statistics, it completely fails to simulate the observed distribution of LKF intersection angles. This deficit can be traced back to the yield curve. The result motivated a dedicated study about the dependence of the intersection angle on the yield curve (Ringeisen et al., 2019), which showed that, with a VP-rheology and classical elliptical yield curve, it is impossible to simulate intersection angles below $60°$ in compression, and hence to reproduce the observed intersection angles. Ringeisen et al. (2019) suggested to use a Mohr-Coulomb yield curve, but also note numerical implementation hurdles. Hutchings et al. (2005) also showed smaller intersection angles using the Mohr-Coulomb yield curve. We hypothesize that the density of LKFs could also increase by improving the simulated intersection angle, because with sharper intersection angles more LKFs can be accommodated in the same area.

LKFs also affect the thermodynamic component of the sea ice model. Once the sea ice cover is opened in a resolved lead, new ice growth is initiated. Koldunov et al. (2019) found that the wintertime sea ice volume increases with increasing number of resolved features. In summertime, ice-ocean interaction along the boundaries of smaller floes accelerate the melting of the ice cover (Horvat et al., 2016), which leads to a lower sea ice volume. Locally, the heat flux and the freshwater fluxes associated with ice melt and growth in leads may generate horizontal gradients and submesoscale variability (Horvat et al., 2016; Manucharyan and Thompson, 2017). There is no feedback of LKFs to drag in the model. As opposed to, for example, Castellani et al. (2018); Tsamados et al. (2014), the LKF density is not a sub grid scale parameterisation, so that previous parameterization of drag as a function of LKF density cannot used. Resolving LKFs allows for a drag parameterisation (e.g., Lüpkes and Gryanik, 2015) that uses information about the free-board and characteristic length scale of floes from the simulated sea ice fields.

## 6  Conclusion

The LKF statistics in this paper provide valuable information about which characteristics of LKFs are reproduced by the model and which modifications to the model are necessary to further improve these simulations. The model simulations, especially the one with an ITD, have LKF fields that are in remarkable agreement with satellite observations from RGPS. This reproduction of realistic deformation features is the prerequisite for regional climate studies that directly resolve atmosphere-ocean interaction

processes along leads. In general, our model configuration could be used to predict deformation features in the ice. When the orientation of leads is of special interest, for example for navigation, modifications to the rheology seem in place to obtain more realistic intersection angles.

So far, scaling analyses are the main tool found in the literature to evaluate lead-permitting sea-ice models and provide insight into the material properties that govern ice dynamics. With our simulations it becomes clear that these analyses cannot discriminate between significantly different model physics in Pan-Arctic simulations (i.e., by comparing our scaling analysis results to Rampal et al., 2019). This suggests that in these scaling analyses the effect of different physics is confounded by external factors such as wind and ocean forcing or interactions with coastlines. Hence, we propose idealized experiments (e.g. as in Dansereau et al., 2016; Weiss and Dansereau, 2017) to study and isolate individual mechanical properties. For the evaluation of deformation features in Pan-Arctic simulations, the direct comparison of these features should be the first choice.

We find that the computed spatio-temporal scaling exponents are mainly linked to the number of LKFs, whereas other direct inferences on other LKF properties are not obvious. For example, high temporal scaling exponents imply high intermittency, but we find that the simulation with higher scaling exponents also tends to have longer LKF lifetimes, which at first glance suggests lower intermittency. Therefore, we do not see our new method as a substitute for existing scaling analyses, but as a complement. The presented LKF statistics offer the opportunity to directly evaluate simulated deformation features. A scaling analysis tests for the material properties of ice dynamics. The decision about an appropriate metric or a combination of both metrics will always depend on the application.

*Code and data availability.* The code of the LKF detection and tracking algorithm is available on github: https://github.com/nhutter/lkf_tools.git (Hutter, 2019a). The RGPS LKF data set is available on PANGAEA: Hutter et al. (2019, https://doi.org/10.1594/PANGAEA.898114). The LKF data sets of both simulations are available on PANGAEA (Hutter, 2019b, c).

## Appendix A:  Modeled sea ice volume and extent

Neither model configuration is tuned to reproduce the observed ice distributions due to limited computational resources. As a consequence, neither simulation reproduces the observed sea ice volume and extent in all detail, but agree in the observed general trend of sea ice retreat (Fig. A1). The lower trend of sea ice volume in the ITD simulations was already reported for the coarse resolution model configuration (Ungermann and Losch, 2018), from which we obtained the ITD specific parameters. Both simulations overestimate the seasonal cycle of sea ice volume, which we attributed partly to the 0-layer thermodynamics used in the simulation and to the effect of resolved leads in the simulation. In wintertime, open-ocean exposed in leads allows further ice growth even for an ice covered Arctic Ocean. In summertime, ice-ocean interaction along the edges of smaller floes accelerate the melting of the ice cover (Horvat et al., 2016). Both model simulations underestimate the maximum sea ice extent, which we attribute to the atmospheric forcing, as both simulations show agree in this underestimation. Improving the agreement with these large scale observations requires a dedicated tuning study, which is beyond the scope of this paper and also beyond

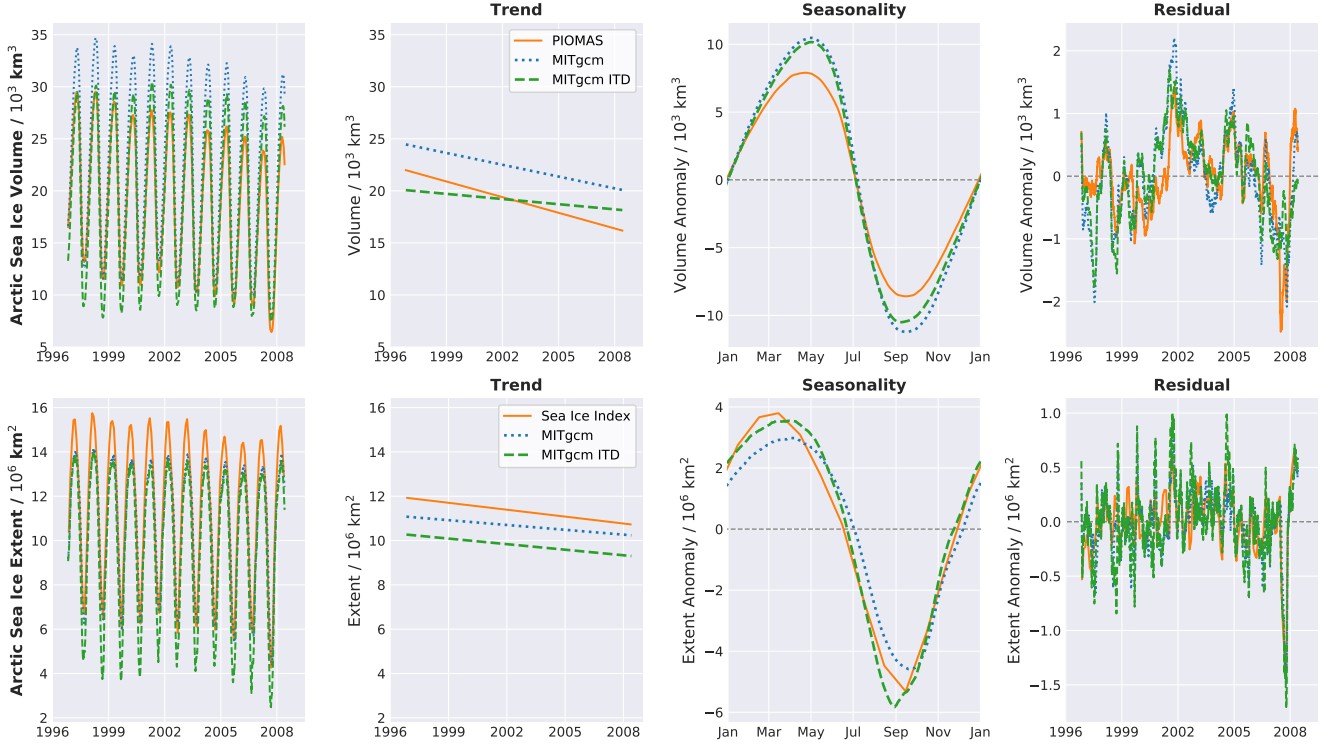

**Figure A1.** (top row) Comparison of Arctic sea ice volume in both model simulations used in our study to the PIOMAS model given as a time series over the entire RGPS period (1996 to 2008) and separated into a linear trend, seasonality and residual. (lower row) Same as upper row but for the Arctic sea ice extent from NSIDC. For a full description of the plots see Ungermann and Losch (2018)

the computer resources available to us. Both simulations agree with observed sea-ice volume and extent reasonably well from November to April, when RGPS data are available, so that we can assume that the model-data differences in summertime are not essential to our analysis.

*Author contributions.* NH performed both model simulations and derived the LKFs data-sets of the simulations. NH developed and imple-
5  mented the LKF statistics and the scaling analysis. NH and ML analyzed and discussed the results of the LKF statistics. NH prepared the manuscript with contributions from ML.

*Competing interests.* The authors declare that they have no conflict of interest.

*Acknowledgements.* We thank Mischa Ungermann for his help setting up the high resolution ITD simulations. We thank Amélie Bouchat for the discussions on scaling analyses that produced valueable input for the manuscript. We are grateful for Lars Kaleschke's comments and discussion on the manuscript.

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
