# Peer review of "Feature-based comparison of sea-ice deformation in lead-permitting sea-ice simulations"

_The Cryosphere, 2019_

## Referee Comment (RC1) · Anonymous Referee #1 · 13 Aug 2019

Review of "Feature-based comparison of sea-ice deformation in lead-resolving sea-ice simulations" by Nils Hutter and Martin Losch

**General comments**

This manuscript presents a set of analyses for feature-based comparison of sea ice deformation. The authors address a useful evaluation, which has been required for some time. The comparison includes a) detecting Linear Kinematic Features (LKF) and b) measuring some of their geometrical characteristics. The sea ice deformation of high-resolution Arctic simulations and an LKF data-set derived from the RARDARSAT Geophysical Processor System (RGPS) are used. The manuscript contains information about the lifetimes and growth rates of LKFs detected from RGPS. The authors suggest the feature-based comparison as an effective substitution for the scaling analysis.

Although the feature-based comparison is well suited for the journal, the used algorithms, and methodology need to be further clarified before the manuscript paper can be accepted for publication. In addition, it is necessary that the objectives of the research be clearly described. Furthermore, the text is wordy and the writing misses conciseness. Authors should clearly describe the originality of their methods in relation to the published ones and avoid reporting style. I outlined some major points, which I would like the authors to consider. I would strongly encourage them to conduct the suggested analyses. It would be constructive if the literature review could contain all relevant researches and projects.

**Major Comments**

1- There is ambiguity in recognizing the novelty of this study. In addition, the manuscript does not explain explicitly the points of providing such comparison. Thus, I inferred that either the authors aim to introduce a new framework for evaluating the numerical results or try to assess the performance of visco-plastic rheology used in a very high-resolution experiment. I review the manuscript for both aspects as follows. The authors are highly encouraged to consider them for the revision.

    a) If the manuscript aims to introduce a new feature-based comparison in sea ice dynamics:
        I- The idea of comparing LKF detected from RGPS and numerical fields and all introduced algorithms in this manuscript have been already published (e.g. Levy et al. 2008, Wang et al. 2016, Hutter et al. 2019, Linow and Dierking 2017, Hutter et al. 2018).
        II- Additional analysis is required to show that the skeletons of LKF could represent the spatial characteristics of LKF.
        III- Although a significant part of the paper is devoted to explaining detecting LKF as an object-based approach, the authors avoid using an object-based comparison. The argument stating that a direct comparison should be avoided due to chaotic dynamics of sea ice is not sufficiently convincing. Is the evaluation of the detecting algorithm (section 3.2, Hutter et al. 2019) not an object-based comparison? Object-based verification of precipitation (e.g. Wernli et al. 2008) is an example that is

applied for chaotic fields. Authors are encouraged to benefit from the advantage of the introduced object-based detecting algorithm.

IV- Again, due to the lack of specific question for the comparison, I assume here that the main goal of developing a comparison framework is assessing the performance of the sea ice models using visco-plastic rheology in a very high-resolution configuration. For such simulations, distribution of LKF (leads in this case, which are controlled by tuning the ice strength parameterisations) is important. However, an analysis, which measures the spatial distribution of LKF, is missing. In addition, it is not clear why comparing the intersection angles and computing scaling characteristics are not sufficient. It is useful to know which physical processes or performance of which numerical schemes are linked to the number of LKF.

V- The authors state that computing scaling characteristics is an old approach and is not appropriate for evaluating the simulated deformation features (line 20 of the second page). Nevertheless, a significant part of the paper is devoted to explaining the scaling analyses and their results. The does not establish useful links between scaling characteristics, and spatial characteristics of LKF. Furthermore, the results of scaling analyses in this paper do not provide new insights into the sea ice dynamics. I suggest removing all these sections unless the authors could emphasize on the positive contribution of the scaling analyses. In this case, I encourage the authors to apply spectral analysis that might be more appropriate for computing the scaling properties of sea-ice deformation (Hutching et al. 2011).

b) If the purpose of the evaluation is an assessment of a specific configuration of the sea ice model.

1- The horizontal resolution of the coupled sea ice-ocean model is pushed to high resolution (~2 km) to resolve much more LKF. However, to reduce the computational costs, the oceanic component of the model has only 16 vertical layers. The authors argued that such configuration is rational since the main purpose of the study is focusing on sea ice processes. In contrast, other configurations of coupled ocean-sea ice models use much more vertical layers to resolve the halocline circulation in the Arctic. For example, Spall (2013) used 30 layers with 50 m thickness on the upper 500 meters and the configuration of Mu et al. 2018 has 50 vertical layers. In addition to resolving the halocline circulation, it is well understood that the number of vertical layers might affect vertical mixing. Liang and Losch (2018) show that vertical mixing affects the vertical heat and salinity exchange. Consequently, they influence directly the sea ice states such as concentration and thickness. Thus, the formation, density, number and all spatial characteristics of LKF might be affected. Thus, the authors should conduct the following analyses.

i-      Discuss whether the current configuration could resolve the corresponding oceanic circulations or not. What are their driving force and their temporal and spatial scale? What type of mixing parameterization is used?

ii-     Compare the main characteristics of the sea ice in the current configuration with the sea ice state of a configuration with comparable horizontal resolution and more vertical layers and/or with any available product, e.g. EUMETSAT OSI SAF for ice concentration. To perform a reasonable comparison, In my opinion, the authors should provide the contours of ice thickness and sea ice strength for sea ice concentration more than 50 % and 85 % for all 12 months of the year.

2-  According to the first comment, the title of the manuscript is very general.

3-  Two different sea-ice simulations are performed. The manuscript does not explain the scientific reason for designing these two simulations. Thus, it is difficult to evaluate the selected comparison methods.

4-  It is argued (Section 4.2.2) that the shape of LKF is scale invariance. This statement is rather subjective. Hutter et al. (2019) showed the LKF detection algorithm terminates detecting LKF when there is a directional change compared to the orientation of the last 5 pixels. Further, they introduced a new starting point. In addition, closed contours are first divided into several segments. The probability that the reconnecting algorithm combines such segmented features is thus questionable.  It means that the algorithm might not be able to detect linear features with high curvature. Overall, I speculate the introduced "number and length of LKF" are not truly spatial features of the LKF and are more and less subjective quantities.

5-  The enhanced horizontal resolution does not necessarily increase the prediction skill (e.g. Mass 2002). A fair analysis discussing position error, double penalty, etc is missing. The analysis should show that the 2 km is a rational horizontal resolution. When the horizontal resolution increases, the objective verification scores might be degraded, although more useful information on a smaller scale is generated. Are deformation of both simulations interpolated into a similar 12.5 km grid? If so why is high-resolution simulation necessary for explaining a new comparison approach?

6-  It is very practical if the authors could tell that how many operators did repeat the seven visual detections of the LKF within one single RGPS image (Linow and Dierking, 2017). To better understand the optimization in detecting LKF explained by Hutter et al. (2019), the evaluation of section (3.2) is highly recommended to be revisited. Try to compare again the uncertainty of the LKF detecting algorithms using a quantification mechanism so that they were comparable with the intrinsic accuracy of the hand-picked lines (Linow and Dierking, 2017).

**References**

Wernli H, Paulat M, Hagen M, Frei C. SAL—A novel quality measure for the verification of quantitative precipitation forecasts. Monthly Weather Review. 2008 Nov; 136(11):4470-87.

Spall MA. On the circulation of Atlantic Water in the Arctic Ocean. Journal of Physical Oceanography. 2013 Nov;43(11):2352-71.

Mu L, Losch M, Yang Q, Ricker R, Losa SN, Nerger L. Arctic-Wide Sea Ice Thickness Estimates From Combining Satellite Remote Sensing Data and a Dynamic Ice-Ocean Model with Data Assimilation During the CryoSat-2 Period. Journal of Geophysical Research: Oceans. 2018 Nov;123(11):7763-80.

Hutchings JK, Roberts A, Geiger CA, Richter-Menge J. Spatial and temporal characterization of sea-ice deformation. Annals of Glaciology. 2011;52(57):360-8.

Hutter N, Zampieri L, Losch M. Leads and ridges in Arctic sea ice from RGPS data and a new tracking algorithm. The Cryosphere. 2019 Feb 20;13(2):627-45.

Linow S, Dierking W. Object-based detection of Linear Kinematic Features in sea ice. Remote Sensing. 2017 May;9(5):493.

Liang X, Losch M. On the Effects of Increased Vertical Mixing on the Arctic Ocean and Sea Ice. Journal of Geophysical Research: Oceans. 2018 Dec;123(12):9266-82.

Mass CF, Ovens D, Westrick K, Colle BA. Does increasing horizontal resolution produce more skillful forecasts? The results of two years of real-time numerical weather prediction over the Pacific Northwest. Bulletin of the American Meteorological Society. 2002 Mar;83(3):407-30.

Hutter N, Losch M, Menemenlis D. Scaling properties of arctic sea ice deformation in a high-resolution viscous-plastic sea ice model and in satellite observations. Journal of Geophysical Research: Oceans. 2018 Jan;123(1):672-87.

Wang Q, Danilov S, Jung T, Kaleschke L, Wernecke A. Sea ice leads in the Arctic Ocean: Model assessment, interannual variability and trends. Geophysical Research Letters. 2016 Jul 16;43(13):7019-27.

Levy G, Coon M, Nguyen G, Sulsky D. Metrics for evaluating linear features. Geophysical Research Letters. 2008 Nov;35(21).

---

## Referee Comment (RC2) · Anonymous Referee #2 · 19 Sep 2019

General comments:

The authors applied some newly developed tracking algorithms for Linear Kinematics Features (LKF) presented in a recent study by the same authors to two model simulations and RGPS data. This approach allows for direct comparison of various metrics of LKF, namely the density, orientation, length, curvature, intersection angles, persistence and growth rates. This study represents a sophisticated assessment of a model's dynamical features. The presentation is clear, the model realism (in terms of those features) convincing, and some interesting results with obvious physical and operational applications. This paper also offers a contribution to a question that has been debated

repeatedly in the community as to the ability for the VP rheologies (and derivatives) to capture the power law distributions seen in the satellite observations. I find that the work is sufficient to justify publication in this journal provided that some of the major issue listed below are addressed.

1) Model tuning. How much of the results are the results of parameter tuning? The authors should make it much clearer if these two model configurations are their standard model simulations and if there was a tuning procedure to obtain such realistic fits to the observations. Additionally, has this tuning been to the detriment of other characteristics of the model (I..e thermodynamic characteristics, sea ice concentration, thickness and velocity). A supplementary plot showing how both models perform with regard to these essential sea ice metrics would be welcome. For example it would not be satisfactory to achieve better fit to the dynamics features discussed in the paper to the detriment of this more standard and important features of the sea ice cover.

2) References to the literature. While some section are well documented, I find other section do not do justice to previous authors who have worked on this theme. Besides the historical studies by Hibler, Coon, Pritchard, Gray, etc the authors also omit more recent work on the anisotropic rheology of Tsamados et al., Tremblay et al, Lemieux, etc. . .

3) Model resolution and forcing dependence of the results. The author present two model runs that differ in their ITD representation (without going too much into the detail of their difference) but fail to present a fair assessment of the sensitivity of their results to the model spatial (and temporal) resolution as well as to the forcing applied. Some definitions (overlap, persistence, etc. . .) are bound to be sensitive to the grid resolution and it would be useful to get a sense of this. An additional, difficult, question that is eluded is the degree of localisation of the LKF in this model. Indeed one crucial quantity of interest of these LKF is their width but the authors fail to discuss that point entirely.

4) Coupling of dynamics with other parts of the model. The authors treat the problem

and the LKFs as if they are completely separate from other components of their model. They make a brief reference to the ridging scheme and drag coefficients but fail to discuss further how modification of LKFs features could couple to other parts of the model.

5) Final major issue that this study uncovered is that the VP rheology fails to capture the intersection angles as they are observed in the observations. This is an important negative result but others have studied these angles before and should be referenced (Hibler, Hutchings, Pritchard, Grey, Ukita, Heorton, ...etc).

Specific comments:

Abstract

P1L6: power law distribution better. Not all power distribution are multi-fractal in nature. P1L9: not an ITD simulation but a sea ice simulation with an ITD parameterization P1L17: addressed

1 Introduction

P2L4: rephrase P2L22: you mean individually? P2L29: one of which P2L34: rephrase P3L2: outline?

2 Methods 2.1 LKF detection and tracking algorithms 2.2 RGPS LKF-dataset 2.3 Model simulations 2.3.1 Model configurations

P3L28: Some have argued that power law is in the forcing? How sensitive are your results to the spatio-temporal lenghtscales of the atmo/ocean forcing? P4L8: we branch -> meaning? P4L13: justify this choice. Cite Landy et al, 2019 P4L19: bold not a good idea. i suggest run_ITD run_noITD

2.3.2 Sampling and LKF extraction

P4L32: what boundary? P5L9: contradicts power law distribution and localisation

2.4 Spatio-temporal scaling analysis

P5L15: and Weiss et al, 2018 for space-time power laws P5L29: this is not clear here
and some repeat of earlier paper might be needed P6L9: than ...(L<L0/2) 2.5 Irregular
temporal sampling of RGPS

P6L19: unclear sentences. What two streams are you referring to in this sentence?
P6L33: Brief algorithm schematic needed in appendix or clear reference to previous
paper, section etc...

3 Scaling in sea-ice deformation

P7L4: General comment: it would be good to know what tuning you have undergone to
achieve such a good fit with the observations. P7L5: decreases P7L16: Can you also
check the space-time scaling as discussed in Weiss et al, 2018 P8L11: Not clear if it is
not good in this study or in Rampal's. Rephrase P8L16: how does this link with power
law exponents? Explain P9L2: This is slightly too strong as they were developed also
to represent some physical characteristics (i.e. stress redistribution...)

4 LKF statistics 4.1 Pan-Arctic distribution of LKFs 4.1.1 Number of LKFs

P10L1:Important consideration is how the results presented below scale with model
resolution but also with spatio-temporal scales of the forcing fields. P10L9: does not
seem significant and also raises questions as to how LKFs are detected in a changing
Arctic P10L22: any suggestions as to why? Generally little elements of physical expla-
nations of the results are given. P10L33: Not clear to me how this relative density is
calculated (what unit?) and how you can compare it to MODIS or CS2 information. For
CS2 please also cite recent paper by Horvat et al, 2019.

4.1.2 LKF density

P13L17: Another possibility is that some of these features are ocean driven
(geostrophic current or Eddies). There is extensive recent literature on this in this
region,

**4.1.3 LKF orientation**

P14 Figure 6: over what period? Season? Specify in caption

**4.2 Spatial LKF properties 4.2.1 LKF length**

P15L7: good review P15L12: So not clear what method you use to measure LKF lengths

**4.2.2 LKF curvature 4.2.3 LKF intersection angles**

P16L29: Cite also study by Hibler and Hutchings, 2004, + several studies by Wilchinsky + Feltham + Tsamados + Heorton on anisotropic rheology with prescribed diamond shaped floes. Tsamados et al, 2013 describes sensitivity to this intersection angle See also papers by Cunningham et al, 1994, Schulson et al, 2006, but also Gray, Coon, Pritchard, Maslowski, Ukita, Moritz... P17 figure 9: So quite important structural difference of the model with reality here. P18L5: angles

**4.3 Temporal evolution of LKFs 4.3.1 LKF persistence**

P18L17: define lifetime calculation (algo). How model resolution is this? Do you calculate persistence in a lagrangian or eulerian way? P18L32: Why didn't you estimate similar biases for the other LKFs characteristics discussed earlier?

**4.3.2 LKF growth rates 5 Discussion**

P22L19: or indirectly in the anisotropic rheologies of Tsamados et al, 2013 via the additional dynamics on the order parameter controlling the degree of anisotropy P22L32: see also Heorton et al, 2019

**6 Conclusion**

---

## Author Comment (AC1) · 18 Oct 2019

**Answers to the reviewers #1**

General comments

This manuscript presents a set of analyses for feature-based comparison of sea ice deformation. The authors address a useful evaluation, which has been required for some time. The comparison includes a) detecting Linear Kinematic Features (LKF) and b) measuring some of their geometrical characteristics. The sea ice deformation of high-resolution Arctic simulations and an LKF data-set derived from the RARDARSAT Geophysical Processor System (RGPS) are used. The manuscript contains information about the lifetimes and growth rates of LKFs detected from RGPS. The authors suggest the feature-based comparison as an effective substitution for the scaling analysis.

Although the feature-based comparison is well suited for the journal, the used algorithms, and methodology need to be further clarified before the manuscript paper can be accepted for publication. In addition, it is necessary that the objectives of the research be clearly described. Furthermore, the text is wordy and the writing misses conciseness. Authors should clearly describe the originality of their methods in relation to the published ones and avoid reporting style. I outlined some major points, which I would like the authors to consider. I would strongly encourage them to conduct the suggested analyses. It would be constructive if the literature review could contain all relevant researches and projects.

We thank the anonymous referee #2 for his review. Please find our detailed answers below.

We would like to point out that we do not recommend to replace the scaling analysis, rather than complement it.

Major Comments

1. There is ambiguity in recognizing the novelty of this study. In addition, the manuscript does not explain explicitly the points of providing such comparison. Thus, I inferred that either the authors aim to introduce a new framework for evaluating the numerical results or try to assess the performance of visco-plastic rheology used in a very high-resolution experiment. I review the manuscript for both aspects as follows. The authors are highly encouraged to consider them for the revision.

The manuscript describes the application of a recently introduced method (Hutter et al., 2019) to a comparison between satellite-based remote sensing data and two numerical model simulations. In doing so, we introduce a new evaluation frame-work and use the two simulations as examples. Thereby, we indirectly also assess the performance of these simulations. We do not see how these two aspects could be split and, therefore, clarified the objective of our study:

The objective of this paper is to establish a feature-based evaluation of sea-ice deformation in lead-resolving sea ice simu- lations. We apply the LKF detection and tracking algorithm of Hutter et al. (2019) to two different sea-ice simulations with a horizontal grid-spacing of 2 km, of which one uses an Ice Thickness Distribution (ITD). We compare the extracted LKFs to an LKF data-set derived from RADARSAT Geophysical Processor System (RGPS) deformation data (Hutter et al., 2019) with respect to their Pan-Arctic distribution (density and orientation), their spatial properties (length, curvature, and intersection angle), and their temporal characteristics (persistence and growth rates). **In addition, we test which conclusions about the properties of LKFs can be drawn from a spatio-temporal scaling analysis of sea ice deformation (following, e.g.**

**Rampal et al., 2016; Hutter et al., 2018).** By analyzing two different model simulations, we study how changes to the model physics, in our case the explicit ridging processes in an ITD model, affect the simulated LKFs, and how the different analysis methods pick up that difference. **With our analysis we test whether the ice strength parameterization of the ITD model, which mainly depends on the thinner ice classes, accelerates lead formation by a faster feedback between deformation, ice thickness, and ice strength as suggested in Hutter et al. (2018).**

a. If the manuscript aims to introduce a new feature-based comparison in sea ice dynamics:
    i. The idea of comparing LKF detected from RGPS and numerical fields and all introduced algorithms in this manuscript have been already published (e.g. Levy et al. 2008, Wang et al. 2016, Hutter et al. 2019, Linow and Dierking 2017, Hutter et al. 2018).

    Other studies have addressed the question how to compare LKFs in observations and simulations: Coon et al. (2007), Levy et al. (2008), and Mohammadi-Aragh (2018) presented metrics that provide one score that quantifies the agreement of the entire field of LKFs. While such comparisons can be added to a cost function in model tuning easily or applied to assess forecast skill, these metrics do not provide insights into the characteristics of LKFs fields, like spatial distribution and persistence, that are important to model the interaction of atmosphere, ocean, and ice along LKFs. Wang et al. (2016) compared lead densities in a model to satellite observations, but no further characteristics, especially no temporal, are deduced.

    We apply the recently introduced method (Hutter et al., 2019, extension of Linow & Dierking, 2017) to detect LKFs in two model simulation and in RGPS. With this analysis, we can explore spatial and temporal characteristics in detail. To our knowledge, such an extensive analysis has not been done before.

    We added the following sentence to make reference to these earlier studies:

    Various metrics for evaluating LKFs as discontinuities in the deformation fields have been suggested, but they all provide only a summary of agreement with a reference in a single score for the entire LKF field (Coon et al., 2007; Levy et al., 2008; Mohammadi-Aragh et al., 2018).

    ii. Additional analysis is required to show that the skeletons of LKF could represent the spatial characteristics of LKF.

    Kwok (2001) first defined LKFs: "*Quasi-linear features of the scale of kilometers to hundreds of kilometers can be observed in the high-resolution deformation fields of the sea ice cover ... They appear as sharp discontinuities separating regions of uniform ice motion. … Here, we refer to them as linear kinematic features (LKFs)."* This definition states that LKFs are quasi-linear and their length is considerably larger than their width. This makes an abstraction of these features to skeletons possible to describe their overall shape such as length,

curvature, orientation, etc. (Banfield, 1992; Van Dyne and Tsatsoulis, 1993; Van Dyne et al., 1998, Linow & Dierking, 2017, Hutter et al., 2019). Only the information about the width of the LKFs is lost by using the skeletons. However, the Lagrangian deformation data with a resolution of 12.5km used in our study does not allow to reliably retrieve the width of an LKF anyways. Thus, no information is lost by representing LKFs by their skeletons.

iii. Although a significant part of the paper is devoted to explaining detecting LKF as an object-based approach, the authors avoid using an object-based comparison. The argument stating that a direct comparison should be avoided due to chaotic dynamics of sea ice is not sufficiently convincing. Is the evaluation of the detecting algorithm (section 3.2, Hutter et al. 2019) not an object-based comparison? Object- based verification of precipitation (e.g. Wernli et al. 2008) is an example that is applied for chaotic fields. Authors are encouraged to benefit from the advantage of the introduced object-based detecting algorithm.

We avoid a direct comparison of deformation features, because a perfect initialisation of the model would be needed for such a comparison due to chaotic dynamics of sea ice. In numerical weather forecasting, sophisticated data assimilation methods are applied to obtain an optimal model initialisation, which makes object-based comparison of short-term precipitation patterns possible. In case of sea ice simulations, knowledge about initial weaknesses in the ice are needed to predict short-term sea-ice deformation and evaluate such predictions with a object-bases one-by-one comparison. In our study, we focus on long-term sea ice simulations without data assimilation and therefore are not able to compare deformation features observed from satellite and modeles one-by-one, but study long-term statistics of both fields.

Note, that in the mentioned object-based comparison in the method evaluation (Section 3.2, Hutter et al., 2019) different algorithms are applied to the same input data, which makes this comparison possible. However, this is completely different from comparing satellite observations to model output.

We changed the text to make the importance of model initialisation more clear: The emergence of deformation features, which can be identified as leads and pressure ridges, calls for a proper evaluation of model simulations against observations. This is challenging because ice mechanics are non-linear and chaotic. A direct comparison of deformation fields bears similar issues as comparing eddy resolving ocean model simulations to high-resolution satellite observations (Mourre et al., 2018). Therefore, it should not be attempted if accurate initial conditions (e.g. obtained by data assimilation) are not available. Still, proper LKF characteristics in sea ice models are important in the context of Arctic climate.

iv. Again, due to the lack of specific question for the comparison, I assume here that the main goal of developing a comparison framework is assessing the

performance of the sea ice models using visco-plastic rheology in a very high-resolution configuration.

The comparison framework presented in our study is not limited to sea ice models using the VP rheology, but can be applied to any simulation that resolve LKFs as localized deformation rates. To present this method, we use two very high resolution VP simulations.

For such simulations, distribution of LKF (leads in this case, which are controlled by tuning the ice strength parameterisations) is important. However, an analysis, which measures the spatial distribution of LKF, is missing. In addition, it is not clear why comparing the intersection angles and computing scaling characteristics are not sufficient.

The reviewer argues in the previous statement that the distribution of LKFs is important. We agree, and we show the spatial distribution of LKFs in Section 4.1.2 "LKF density" and discuss the temporal distribution of LKFs in Section 4.1.1 "Number of LKFs". Scaling characteristics and intersection angles of LKF provide useful insight into the deformation physics that can be used to modify the rheology. In our study, however, we present two simulations that show similar scaling characteristics and intersection angles, but different number of LKFs and different LKF lifetimes. These differences will have an impact once such sea ice models are used in coupled climate modeling. The point of the paper is that we should not restrict ourselves to very few diagnostics.

It is useful to know which physical processes or performance of which numerical schemes are linked to the number of LKF.

In the study, we show that applying an ITD model increases the number of simulated LKFs (see Sections 4.1.1, 4.1.2. and 5). In Section 5, we discuss that the implementation of a damage parameter will also have a positive effect on the number of simulated LKFs.

v.  The authors state that computing scaling characteristics is an old approach and is not appropriate for evaluating the simulated deformation features (line 20 of the second page). Nevertheless, a significant part of the paper is devoted to explaining the scaling analyses and their results. The does not establish useful links between scaling characteristics, and spatial characteristics of LKF. Furthermore, the results of scaling analyses in this paper do not provide new insights into the sea ice dynamics. I suggest removing all these sections unless the authors could emphasize on the positive contribution of the scaling analyses. In this case, I encourage the authors to apply spectral analysis that might be more appropriate for computing the scaling properties of sea-ice deformation (Hutching et al. 2011).

The phrase "for evaluating the simulated deformation features themselves" is misleading and we removed it. We never meant to say that the scaling analysis is

"old" and "not appropriate" (and we don't use these exact words anywhere). We mean "individual features" and rephrase the sentence to

"While scaling characteristics give some insight into the underlying material properties of sea ice, their interpretation with respect to individual deformation features is not straightforward (Bouchat and Tremblay, 2017; Hutter et al., 2018). ... A comprehensive description of individual deformation features requires their detection to extract statistics such as density, orientation, intersection angle, and persistence."

The scaling analysis is interesting here, because it does provide useful information about LKFs, but it also provides a direct comparison to previous work (e.g. Girard et al., 2009; Rampal et al. 2016; Rampal et al., 2019). We cannot use it here to differentiate between the two simulations, because both simulations reproduced the observed scaling of RGPS data. This is important as failure to do so in previous coarse resolution VP models has been used to argue against their use (e.g. Girard et al 2009). The higher moment analysis is the first successful match of a VP-model to RGPS, to our knowledge.

b. If the purpose of the evaluation is an assessment of a specific configuration of the sea ice model.

  i. The horizontal resolution of the coupled sea ice-ocean model is pushed to high resolution (~2 km) to resolve much more LKF. However, to reduce the computational costs, the oceanic component of the model has only 16 vertical layers. The authors argued that such configuration is rational since the main purpose of the study is focusing on sea ice processes. In contrast, other configurations of coupled ocean-sea ice models use much more vertical layers to resolve the halocline circulation in the Arctic. For example, Spall (2013) used 30 layers with 50 m thickness on the upper 500 meters and the configuration of Mu et al. 2018 has 50 vertical layers. In addition to resolving the halocline circulation, it is well understood that the number of vertical layers might affect vertical mixing. Liang and Losch (2018) show that vertical mixing affects the vertical heat and salinity exchange. Consequently, they influence directly the sea ice states such as concentration and thickness. Thus, the formation, density, number and all spatial characteristics of LKF might be affected. Thus, the authors should conduct the following analyses:

More vertical ocean layers may affect the ice thickness, but the main driver of sea ice thermodynamics is the atmospheric forcing. It is beyond the scope of this manuscript to demonstrate that. In Figure 1 (of this response), we compare the performance of both simulations with respect to sea ice volume and extent. In this comparison period between November to April, both simulations show reasonable sea-ice volume and extent, such that we evaluate the misfit in spring and fall as non-essential to our analysis (For a full discussion see Answers to reviewer #2). In addition, the LKF densities presented for RGPS in Figure 5a (of the manuscript) do not vary strongly between the thin first-year ice and thick multi-year ice. Thus, we assess that the misfit of sea ice volume presented in

Figure 1, does not impact significantly the numbers and distributions of simulated LKFs.

We agree with the reviewer that the vertical resolution is a very important aspect of an ocean model. Especially the circulation of the Atlantic Water depends strongly on vertical mixing and vertical grid spacing (to the extent that it affects numerical mixing in the vertical, e.g. Spall 2013). But we do not (and will never) use the present model configuration with only 16 vertical layers for studies of the Arctic Ocean interior. For sea ice dynamics and thermodynamics, the ocean surface is important (apart from the much more important atmospheric forcing fields, which to 1st order determine the ice extent), because it exerts stress (dynamical forcing) and because it imposes a heat flux (thermodynamic forcing). Our analysis is restricted to the ice covered part (mostly western Arctic in the winter months), where RGPS data are available. In this region and at this time, the impact of vertical mixing on the ice extent and surface temperature is practically independent of changing vertical mixing coefficient over 2 orders of magnitude (at least in the simulations of Liang and Losch 2018, where the grid spacing is 18km). The ice thickness does decrease with more vertical mixing in Liang and Losch (2018), but not by more that 50cm, except for a small area near the Canadian Arctic Archipelago. This is well within the general uncertainty of sea ice thickness estimates, and especially sea ice model biases.

[Figure]

**Figure 1: (top row)** Comparison of Arctic sea ice volume in both model simulations used in our study to the PIOMAS model given as a time series over the entire RGPS period (1996 to 2008) and separated into

a linear trend, seasonality and residual. **(lower row)** Same as upper row but for the Arctic sea ice extent from NSIDC. For a full description of the plots see Ungermann & Losch (2018).

We compared our horizontal surface fields and vertical sections of temperature and salinity through the Central Arctic to previous simulations with 50 layers (as in Mu et al 2018), but a 4km horizontal grid spacing and find that grossly the vertical structure is similar between the models, but indeed the vertical gradients of temperature and salinity in the 16 layer model are not as strong as in the 50 layer model, as expected (Fig. 2 and 3 of this reply). The surface fields, however, are not significantly different (Fig. 4 of this reply). It was important to us that the surface ocean reproduces some of the observed small scale eddy activity as an additional forcing for the sea ice dynamics, and indeed the fronts and eddies in the 2km surface fields are better resolved than in coarser simulations, see attached figures:

[Figure]

**Figure 2**: Temperature transect of the Arctic ocean along prime meridian **(upper)** in the 2km configuration with 16 vertical layers used in our study and **(lower)** in a 4km configuration with 50 vertical layers (Mu et al., 2018).

[Figure]

**Figure 3:** Same as Fig. 1 but showing salinity

[Figure]

**Figure 4:** Sea surface temperature in the Arctic Ocean **(upper)** in the 2km configuration with 16 vertical layers used in our study and **(lower)** in a 4km configuration with 50 vertical layers (Mu et al., 2018).

Please note that there are also realistic sea ice simulations that are coupled to a simple "slab-ocean" with coarse and non-reactive ocean circulation (e.g. Rampal 2016)

1. Discuss whether the current configuration could resolve the corresponding oceanic circulations or not. What are their driving force and their temporal and spatial scale? What type of mixing parameterization is used?

The model configuration in Section 2.3.1 includes grid resolution, bathymetry, lateral boundary conditions, initial conditions, surface forcing (even with time resolution and spatial resolution), and a reference to Nguyen et al (2011) for the configuration of the ocean model (mixing, etc.) Nguyen et al. (2011) focussed on the ocean circulation. We use the same ocean parameters, i.e. the vertical mixing scheme is KPP (Large et al 1994) and a modified Leith (1996) scheme in the horizontal. The ocean circulation will not be the same as in Nguyen et al 2011, because our horizontal resolution is higher, our vertical resolution is lower, our forcing is different, and even our initial conditions are different. However, the ocean is only interesting to the extent to which it drives the ice (mostly dynamics), i.e. the short scale eddy driving stress forcing (see also previous answer). Discussing details of the ocean circulation is well beyond the scope of this paper (assessing the statistics of LKFs in two different sea ice formulations in comparison to RGPS). We don't even find it appropriate in this journal. We added the following statement Section 2.3.1:

The resulting ocean circulation has not been evaluated in detail, but the wind-driven surface circulation is plausible with strong mesoscale activity, the surface temperature and large scale sea ice distribution follow the prescribed surface forcing as expected. The main role of the ocean model is to provide dynamic bottom boundary conditions to the sea ice model.

2. Compare the main characteristics of the sea ice in the current configuration with the sea ice state of a configuration with comparable horizontal resolution and more vertical layers and/or with any available product, e.g. EUMETSAT OSI SAF for ice concentration.

To our knowledge, there are no sea-ice ocean simulations at this very high horizontal resolution publicly available that have more vertical layers and a similar simulation period. Therefore, we use satellite products and reanalysis to assess the performance of the two simulations (Figure 1 of this answer).

To perform a reasonable comparison, In my opinion, the authors should provide the contours of ice thickness and sea ice strength for sea ice concentration more than 50 % and 85 % for all 12 months of the year.

This would mean 96 plots, for which there is no space in the manuscript. Therefore, we prefer to compare Arctic wide properties in the more compact way above.

It's not clear, what we should compare ice thickness and ice strength to, because we do have either observational data available for this comparison, nor numerical model simulations at similar resolution.

2. According to the first comment, the title of the manuscript is very general.

The title of our manuscript is "Feature-based comparison of sea-ice deformation in lead-resolving sea-ice simulations". We do not agree that it is too general: In our study we compare the spatial and temporal characteristics of deformation features present as LKFs in deformation fields. This comparison is based on the detection of single LKFs, which is summarized in the title as "feature-based comparison of sea-ice deformation". We compare deformation features extracted from satellite observations (RGPS) and two sea ice simulations that resolve leads by using a very high grid spacing and the VP rheology. These simulations are referred to as "lead-resolving sea-ice simulations".

3. Two different sea-ice simulations are performed. The manuscript does not explain the scientific reason for designing these two simulations. Thus, it is difficult to evaluate the selected comparison methods.

The comment is not clear to us. We use two simulations with different levels of sophistication that are reflected mainly in the ice strength parameterisation. We show, how they compare to remote sensing data and draw some conclusions about the dynamics. To be more clear, we added a sentence to the objectives of the paper to highlight why we chose to change the ITD and the ice strength parameterisation, which is also further discussed in Section 5.:
With our analysis we test whether the ice strength parameterization of the ITD model, which mainly depends on the thinner ice classes, accelerates lead formation by a faster feedback between deformation, ice thickness, and ice strength as suggested in Hutter et al. (2018).

4. It is argued (Section 4.2.2) that the shape of LKF is scale invariant. This statement is rather subjective. Hutter et al. (2019) showed the LKF detection algorithm terminates detecting LKF when there is a directional change compared to the orientation of the last 5 pixels. Further, they introduced a new starting point. In addition, closed contours are first divided into several segments. The probability that the reconnecting algorithm combines such segmented features is thus questionable. It means that the algorithm might not be able to detect linear features with high curvature.

The reconnection instance in the LKF detection algorithms prefers to reconnect segments that show little difference in the orientation in accordance to the definition of LKFs given above. Nevertheless, it will reconnect segments up to an upper limit of 35° for the difference in the orientation, if no better matching segments are present. By consecutively reconnecting small segments, LKFs with higher curvature can be detected (we note that this will only happen for larger features that span multiple pixels). Linow & Dierking (2017) studied the curvature of LKFs

from hand-picked data and found low curvatures with close to linear LKFs. This shows that are no high curvature LKFs that the detection algorithm could miss.

Overall, I speculate the introduced "number and length of LKF" are not truly spatial features of the LKF and are more and less subjective quantities.

We do not understand what this statement is based on, as it is not linked to the curvature discussed above. We require more information to give a better answer.

5. The enhanced horizontal resolution does not necessarily increase the prediction skill (e.g. Mass 2002). A fair analysis discussing position error, double penalty, etc is missing. The analysis should show that the 2 km is a rational horizontal resolution. When the horizontal resolution increases, the objective verification scores might be degraded, although more useful information on a smaller scale is generated. Are deformation of both simulations interpolated into a similar 12.5 km grid? If so why is high-resolution simulation necessary for explaining a new comparison approach?

Numerical weather prediction as in Mass et al, is a different subject: the dynamical system is completely different (based on Navier-Stokes/Primitive Equations and probably data assimilation); we do not evaluate predictive skill, but dynamical properties of the model. We do not claim, that 2km increases any skill. It just leads to more LKFs. Lower resolutions do not exhibit these features and hence a feature based analysis method could not be applied.

6. It is very practical if the authors could tell that how many operators did repeat the seven visual detections of the LKF within one single RGPS image (Linow and Dierking, 2017). To better understand the optimization in detecting LKF explained by Hutter et al. (2019), the evaluation of section (3.2) is highly recommended to be revisited. Try to compare again the uncertainty of the LKF detecting algorithms using a quantification mechanism so that they were comparable with the intrinsic accuracy of the hand-picked lines (Linow and Dierking, 2017).

All of these comments relate to already published papers with a completed peer review. Repeating previous work is beyond the scope of the manuscript.

**References**

Wernli H, Paulat M, Hagen M, Frei C. SAL—A novel quality measure for the verification of quantitative precipitation forecasts. Monthly Weather Review. 2008 Nov; 136(11):4470-87.

Spall MA. On the circulation of Atlantic Water in the Arctic Ocean. Journal of Physical Oceanography. 2013 Nov;43(11):2352-71.

Mu L, Losch M, Yang Q, Ricker R, Losa SN, Nerger L. Arctic-Wide Sea Ice Thickness Estimates From Combining Satellite Remote Sensing Data and a Dynamic Ice-Ocean Model with Data Assimilation During the CryoSat-2 Period. Journal of Geophysical Research: Oceans. 2018 Nov;123(11):7763-80.

Hutchings JK, Roberts A, Geiger CA, Richter-Menge J. Spatial and temporal characterization of sea-ice deformation. Annals of Glaciology. 2011;52(57):360-8.

Hutter N, Zampieri L, Losch M. Leads and ridges in Arctic sea ice from RGPS data and a new tracking algorithm. The Cryosphere. 2019 Feb 20;13(2):627-45.

Linow S, Dierking W. Object-based detection of Linear Kinematic Features in sea ice. Remote Sensing. 2017 May;9(5):493.

Liang X, Losch M. On the Effects of Increased Vertical Mixing on the Arctic Ocean and Sea Ice. Journal of Geophysical Research: Oceans. 2018 Dec;123(12):9266-82.

Mass CF, Ovens D, Westrick K, Colle BA. Does increasing horizontal resolution produce more skillful forecasts? The results of two years of real-time numerical weather prediction over the Pacific Northwest. Bulletin of the American Meteorological Society. 2002 Mar;83(3):407-30.

Hutter N, Losch M, Menemenlis D. Scaling properties of arctic sea ice deformation in a high-resolution viscous-plastic sea ice model and in satellite observations. Journal of Geophysical Research: Oceans. 2018 Jan;123(1):672-87.

Wang Q, Danilov S, Jung T, Kaleschke L, Wernecke A. Sea ice leads in the Arctic Ocean: Model assessment, interannual variability and trends. Geophysical Research Letters. 2016 Jul 16;43(13):7019-27.

Levy G, Coon M, Nguyen G, Sulsky D. Metrics for evaluating linear features. Geophysical Research Letters. 2008 Nov;35(21).

**References:**

Cunningham, G. F., Kwok, R., and Banfield, J.: Ice lead orientation characteristics in the winter Beaufort Sea, in: Proceedings of IGARSS '94 - 1994 IEEE International Geoscience and Remote Sensing Symposium, vol. 3, pp. 1747–1749 vol.3, https://doi.org/10.1109/IGARSS.1994.399553, 1994.

Haas, C.: Dynamics Versus Thermodynamics: The Sea Ice Thickness Distribution, chap. 4, pp. 113–151, John Wiley Sons, Ltd, https://doi.org/10.1002/9781444317145.ch4, https://onlinelibrary.wiley.com/doi/abs/10.1002/9781444317145.ch4, 2010.

Horvat, C., E. Tziperman, and J.-M. Campin (2016), Interaction of sea ice floe size, ocean eddies, and sea ice melting, Geophys. Res. Lett., 43, 8083–8090, doi:10.1002/2016GL069742.

Horvat, C., Roach, L., Tilling, R., Bitz, C., Fox-Kemper, B., Guider, C., Hill, K., Ridout, A., and Sheperd, A.: Estimating The Sea Ice Floe Size Distribution Using Satellite Altimetry: Theory, Climatology, and Model Comparison, The Cryosphere Discuss., https://doi.org/10.5194/tc-2019-134, in review, 2019.

Hutchings, J. K., Heil, P., and Hibler, W. D.: Modeling Linear Kinematic Features in Sea Ice, Monthly Weather Review, 133, 3481–3497, https://doi.org/10.1175/MWR3045.1, https://doi.org/10.1175/MWR3045.1, 2005.

Hutter, N.: Viscous-plastic sea-ice models at very high resolution, Master's thesis, University of Bremen, Alfred Wegener Institute, Helmholtz Centre for Polar and Marine research, https://doi.org/10013/epic.46129, http://dx.doi.org/10013/epic.46129, 2015

Koldunov, N. V., Danilov, S., Sidorenko, D., Hutter, N., Losch, M., Goessling, H., et al. (2019). Fast EVP solutions in a high-resolution sea ice model. *Journal of Advances in Modeling Earth Systems*, 11, 1269– 1284. https://doi.org/10.1029/2018MS001485

Kwok, R. and Cunningham, G. F.: RADARSAT GEOPHYSICAL PROCESSOR SYSTEM: DATA USER'S HANDBOOK (Version 2.0), 2014.

Large, W. G., McWilliams, J. C., and Doney, S. C. (1994), Oceanic vertical mixing: A review and a model with a nonlocal boundary layer parameterization, *Rev. Geophys.*, 32( 4), 363– 403, doi:10.1029/94RG01872.

Leith, C. E., Stochastic models of chaotic systems, *Physica D.*, *98*, 481-491, doi:10.1016/0167-2789(96)00107-8, 1996.

Levy, G., Coon, M., Nguyen, G., and Sulsky, D.: Metrics for evaluating linear features, Geophysical Research Letters, 35, https://doi.org/10.1029/2008GL035086, https://agupubs.onlinelibrary.wiley.com/doi/abs/10.1029/2008GL035086, 2008.

Lüpkes, C. and Gryanik, V. (2015): Parameterization of drag coefficients over polar sea ice for climate models, Mercator Ocean Quarterly Newsletter - Special Issue, 51 , pp. 29-34.

Menemenlis, D., I. Fukumori, and T. Lee (2005), Using Green's functions to calibrate an ocean general circulation model, Mon. Weather Rev., 133(5), 1224–1240, doi:10.1175/MWR2912.1.

Mohammadi-Aragh, M., Goessling, H. F., Losch, M., Hutter, N., and Jung, T.: Predictability of Arctic sea ice on weather time scales, Scientific reports, 8, 2018.

Schulson, E., Fortt, A., Iliescu, D., and Renshaw, C.: On the role of frictional sliding in the compressive fracture of ice and granite: Terminal vs. post-terminal failure, Acta Materialia, 54, 3923 – 3932, https://doi.org/https://doi.org/10.1016/j.actamat.2006.04.024, http://www.sciencedirect.com/science/article/pii/S1359645406003120, 2006.

Sumata, H., F. Kauker, M. Karcher, and R. Gerdes, 2019: Simultaneous Parameter Optimization of an Arctic Sea Ice–Ocean Model by a Genetic Algorithm. Mon. Wea. Rev.,147, 1899–1926, https://doi.org/10.1175/MWR-D-18-0360.1
Ukita, J., and R. Moritz (1994), Yield curves and flow rules of pack ice, Journal of Geophysical Research-Oceans, 100(C3), 4545–4557.

Tsamados, M., Feltham, D. L., and Wilchinsky, A. V.: Impact of a new anisotropic rheology on simulations of Arctic sea ice, Journal of Geophysical Research: Oceans, 118, 91–107, https://doi.org/10.1029/2012JC007990, https://agupubs.onlinelibrary.wiley.com/doi/abs/10.1029/2012JC007990, 2013.

Ukita, J. and Moritz, R. E.: Yield curves and flow rules of pack ice, Journal of Geophysical Research: Oceans, 100, 4545–4557, https://doi.org/10.1029/94JC02202, https://agupubs.onlinelibrary.wiley.com/doi/abs/10.1029/94JC02202, 1995

Wadhams, P.: Sea ice thickness distribution in the Greenland Sea and Eurasian Basin, May 1987, Journal of Geophysical Research: Oceans, 97, 5331–5348, https://doi.org/10.1029/91JC03137, https://agupubs.onlinelibrary.wiley.com/doi/abs/10.1029/91JC03137, 1992.

Zhao, M., Timmermans, M.-L., Cole, S., Krishfield, R., Proshutinsky, A., and Toole, J.: Characterizing the eddy field in the Arctic Ocean halo- cline, Journal of Geophysical Research: Oceans, 119, 8800–8817,

https://doi.org/10.1002/2014JC010488, https://agupubs.onlinelibrary.
wiley.com/doi/abs/10.1002/2014JC010488, 2014.

---

## Author Comment (AC2) · 18 Oct 2019

**Answers to the reviewers #2**

**General comments:**

The authors applied some newly developed tracking algorithms for Linear Kinematics Features (LKF) presented in a recent study by the same authors to two model simulations and RGPS data. This approach allows for direct comparison of various metrics of LKF, namely the density, orientation, length, curvature, intersection angles, persistence and growth rates. This study represents a sophisticated assessment of a model's dynamical features. The presentation is clear, the model realism (in terms of those features) convincing, and some interesting results with obvious physical and operational applications. This paper also offers a contribution to a question that has been debated repeatedly in the community as to the ability for the VP rheologies (and derivatives) to capture the power law distributions seen in the satellite observations. I find that the work is sufficient to justify publication in this journal provided that some of the major issues listed below are addressed.

We thank the anonymous referee #2 for the thorough review.

1) **Model tuning.** How much of the results are the results of parameter tuning? The authors should make it much clearer if these two model configurations are their standard model simulations and if there was a tuning procedure to obtain such realistic fits to the observations. Additionally, has this tuning been to the detriment of other characteristics of the model (I..e thermodynamic characteristics, sea ice concentration, thickness and velocity). A supplementary plot showing how both models perform with regard to these essential sea ice metrics would be welcome. For example it would not be satisfactory to achieve better fit to the dynamics features discussed in the paper to the detriment of this more standard and important features of the sea ice cover.

The model simulations are very expensive, with respect to the resources available to us, so that we did not tune the simulations, but rather used sea ice and ocean model parameters from a coarse simulation with realistic ice distributions (Nguyen et al., 2011, and ITD specific parameters from Ungermann & Losch, 2018). Objective and automated tuning methods require large repetitions of model simulations for different parameter choices and/or ensemble members (Menemenlis et al., 2005; Nguyen et al., 2011; Massonnet et al., 2014; Ungermann et al., 2017; Sumata et al., 2019), which is currently not feasible for resolutions as high as 2km.

[Figure]

**Figure 1: (top row)** Comparison of Arctic sea ice volume in both model simulations used in our study to the PIOMAS model given as a time series over the entire RGPS period (1996 to 2008) and separated into a linear trend, seasonality and residual. **(lower row)** Same as upper row but for the Arctic sea ice extent from NSIDC. For a full description of the plots see Ungermann & Losch (2018).

As a consequence, both simulations do not reproduce the sea ice volume and extent observed from satellites and reanalysis products in all the details, but agree in the overall trend of sea ice retreat (Fig. 1 of this answer). The lower trend of sea ice volume in the ITD simulations was already reported for the coarse resolution model configuration (Ungermann & Losch, 2018), from which we obtained the ITD specific parameters. The seasonal cycle of sea ice volume is strongly overestimated, which we attributed partly to the 0-layer thermodynamics used in the simulation and to the effect of resolved leads in the simulation. In wintertime, open-ocean is exposed in leads which allows further ice growth even for an ice covered Arctic Ocean. In summertime, ice-ocean interaction along the boundaries of smaller floes accelerate the melting of the ice cover (Horvat et al., 2016). Both model simulations underestimate the maximum sea ice extent, which we attribute to the atmospheric forcing, as both simulations show agree in this underestimation. In conclusion, there is potential to improve the performance of both simulation by tuning model parameters. Due to limited computing resources this task needs to be done in a dedicated study. Our analysis, however, focusses on the RGPS winter coverage, where most data is available from November to April. For this period of the year, both simulations show reasonable sea-ice volume and extent, such that we evaluate the presented drawbacks as non-essential to our analysis.

We added further details to the model description in the manuscript to highlight the fact that both configurations are untuned:

Both model configurations are not tuned to reproduce observed ice distributions due to limited computational resources. Instead, we carried over ocean and sea ice parameters from optimized coarse resolution configurations (Nguyen et al., 2011; Ungermann & Losch, 2018, for ITD specific parameters).

The resulting simulations overestimate the seasonal amplitude of sea ice volume and extent, but their trends are reasonable (not shown). The resulting ocean circulation has not been evaluated in detail, but the wind-driven surface circulation is plausible with strong mesoscale activity, the surface temperature and large scale sea ice distribution follow the prescribed surface forcing as expected. The main role of the ocean model is to provide dynamic bottom boundary conditions to the sea ice model.

And in the discussion section:

The agreement of the scaling analysis and the LKFs statistics between the model simulations and RGPS data may appear almost surprising given that the models have not been tuned at all for these diagnostics. We argue that this model performance is not determined by the large scale distribution of sea ice thickness and concentration, but the plastic model physics. For the plastic physics in VP-models to produce highly intermittent and heterogeneous LKF distributions, high resolution (Spreen et al. 2016, Hutter et al. 2018) and a sufficiently accurate solver (Koldunov et al. 2019) are necessary. As long as there is a quasi-closed ice cover, which is the case where and when the RGPS data are available, the plastic physics will produce localized deformation --- even in idealized configurations (Hutter 2015, Heorteon et al. 2019) --- and the associated statistics.

2) **References to the literature.** While some sections are well documented, I find other sections do not do justice to previous authors who have worked on this theme. Besides the historical studies by Hibler, Coon, Pritchard, Gray, etc the authors also omit more recent work on the anisotropic rheology of Tsamados et al., Tremblay et al, Lemieux, etc…

We agree that some literature was missing in the first draft of the manuscript and added 17 references in the revision. In particular we focussed on the section of the intersection angles highlighted by the reviewer in the major comment 5) and in the minor comments. Please note that studies of Hibler, Coon, Pritchard had been cited in the paper. We now also include reference to anisotropic rheology.

3) **Model resolution and forcing dependence of the results.** The author present two model runs that differ in their ITD representation (without going too much into the detail of their difference) but fail to present a fair assessment of the sensitivity of their results to the model spatial (and temporal) resolution as well as to the forcing applied. Some definitions (overlap, persistence, etc. . .) are bound to be sensitive to the grid resolution and it would be useful to get a sense of this. An additional, difficult, question that is eluded is the degree of localisation of the LKF in this model. Indeed one crucial quantity of interest of these LKF is their width but the authors fail to discuss that point entirely.

The representation of LKFs is model resolution dependent: with lower resolution there are fewer LKFs and deformation rates are less localized, such that the scaling properties deteriorate (e.g. Spreen et al., 2017). The feature-based evaluation presented in this study can only be applied to simulations that explicitly resolve LKFs, which in our experience is only possible for resolutions higher than 5km. This leaves only a small range of model resolution (2 to 5km) for a sensitivity study. Forcing resolution has a similar effect, with higher resolution leads to more localized deformation (Hutter, 2015). With our high resolution configuration it would be possible to test the effect of different atmospheric forcing. Nevertheless, we find that further sensitivity studies are beyond the scope of the manuscript, as the main purpose is to demonstrate a new way to evaluate lead-resolving sea ice simulations. We agree that this

topic is interesting and extensive enough for a dedicated study. We added a paragraph to the Discussions to discuss this effect:

Both model simulations use the same grid and the same atmospheric forcing, which precludes direct inferences of resolution impact on the presented statistics. Here, we comment on expected impact based on previous studies. With increasing horizontal grid spacing, deformation features are more localized and more frequent (Spreen et al., 2016). Thus the number of LKFs, presented in Section 4.1.1, are likely to increase with model resolution along with a decrease in LKF length and growth rates as discussed in the previous paragraph. In idealized experiments, it has been shown that higher spatial resolution of the atmospheric forcing also has the potential to increase the localization of sea-ice deformation (Hutter, 2015). This suggests that also the number of LKFs increase. We speculate, however, that in our simulations this effect is saturated because we already use atmospheric forcing with fairly high resolution (JRA-55, 0.5625°) that resolves most scales associated with the wind.. To our knowledge, there is no study on the impact of temporal resolution on sea ice deformation. We hypothesize that an increased temporal resolution of the forcing will increase the short-term variability in the evolution LKFs, with direct impact on the LKF growth rates and presence of short-lived LKFs. However, please note that the order of this short-term variability is given by the temporal resolution (hours), which is much smaller than the shortest LKF lifetimes regarded in our study (0-3 days).

We do consider in our study that some parameters of the detection algorithm are resolution dependent. Thus these parameters are scaled accordingly to the model resolution as suggested in Hutter et al., 2019. We now made clear that this adjustment is due to the difference in resolution between the simulations and RGPS data:

The parameters used in the detection algorithm are the same as in Hutter et al. (2019, their Tab. 1), where all parameters marked with [b] are scaled to the reduced model resolution by multiplying with a factor of 12.5 km/6.75 km = 1.85 to account for the resolution difference between the simulations and the RGPS data set.

We agree that the persistence of LKFs is sensitive to the temporal resolution of the atmospheric forcing (in our case 3h) and the time step used by the model (in our case 120s) if lifetimes close to these time scales are regarded. Both time scales are much smaller than the upper bound of the lowest bin of LKF lifetime presented (0 to 3 days). Thus, we do not expect any direct effect on the presented distributions of LKFs lifetimes. We speculate that LKF persistence as well as LKF length and LKF density are sensitive to the number of LKFs and thereby indirectly to the model grid spacing. However, we find that dedicated sensitivity studies are beyond the scope of this study.

We do not discuss the width of LKFs, because LKFs presented in this evaluation are derived from deformation data, which do not allow accurate estimates for the width. Due to the Lagrangian nature of the RGPS data-set deformation features that are one pixel wide can have any width up to 12.5km and multiple pixel wide features are most likely a set of leads. We refrain, therefore, from computing LKF width from the detected LKFs and recommend to use higher resolution satellite products to do so.

**4) Coupling of dynamics with other parts of the model.** The authors treat the problem and the LKFs as if they are completely separate from other components of their model. They make a brief reference to

the ridging scheme and drag coefficients but fail to discuss further how modification of LKFs features could couple to other parts of the model.

We discuss extensively how the resolved LKFs impact on the ice strength and deformation. Besides this link to the dynamics of the sea ice model, there is a clear link to the thermodynamic component of the sea ice model. Once the sea ice cover is opened in a resolved lead, new ice growth is initiated. In the time series of the sea ice volume (Figure 1 of this answer) this effect leads to the overestimation of winter ice volume. Koldunov et al. (2019) showed that the sea ice volume in wintertime increases with increasing number of resolved feature by varying parameters of the EVP solver. In summertime, ice-ocean interaction along the boundaries of smaller floes accelerate the melting of the ice cover (Horvat et al., 2016), which leads to a lower sea ice volume. The heat and freshwater fluxes associated with ice melt and growth in leads might affect the ocean model locally.

There is no feedback of LKFs to drag in the model. As opposed to e.g. Castellani et al (2018), Tsamados et al (2014), LKF density is not a sub grid scale parameterisation, so that previous parameterization of drag as a function of LKF density cannot used. One could introduce a drag parameterisation (e.g., Lüpkes & Gryanik, 2015) that uses information about the freeboard and characteristic length scale of floes from the simulated sea ice fields that include resolved leads and floes.

We added the following paragraph to the Discussions:

LKFs also affect the thermodynamic component of the sea ice model. Once the sea ice cover is opened in a resolved lead, new ice growth is initiated. Koldunov et al. (2018) found that the wintertime sea ice volume increases with increasing number of resolved features. In summertime, ice-ocean interaction along the boundaries of smaller floes accelerate the melting of the ice cover (Horvat et al., 2016), which leads to a lower sea ice volume. Locally, the heat flux and the freshwater fluxes associated with ice melt and growth in leads may generate horizontal gradients and submesoscale variability (Horvat et al., 2016; Manucharyan & Thompson, 2017). There is no feedback of LKFs to drag in the model. As opposed to, for example, (Castellani et al., 2018; Tsamados et al., 2014), the LKF density is not a sub grid scale parameterisation, so that previous parameterization of drag as a function of LKF density cannot used. Resolving LKFs allows for a drag parameterisation (e.g., Lüpkes & Gryanik, 2015) that uses information about the free-board and characteristic length scale of floes from the simulated sea ice fields.

5) Final major issue that this study uncovered is that the VP rheology fails to capture the intersection angles as they are observed in the observations. This is an important negative result but others have studied these angles before and should be referenced (Hibler, Hutchings, Pritchard, Grey, Ukita, Heorton, . . .etc).

We agree, and now also make more references to previous work. Details are given to the specific comments about Section 4.2.3 LKF intersection angles.

**Specific comments:**

P1L6: power law distribution better. Not all power distribution are multi-fractal in nature.

We agree that not all power-law distributions are multi-fractal, but in Section 3 we show that the modeled and observed sea ice deformation shows multi-fractal properties. This information would be lost by replacing it with "power law distributions", so that we keep "multi-fractal".

P1L9: not an ITD simulation but a sea ice simulation with an ITD parameterization
Changed accordingly.

P1L17: addressed
Changed accordingly.

P2L4: rephrase
We removed the sentence.

P2L22: you mean individually?
We changed the sentence to:
While scaling characteristics give some insight into the underlying material properties of sea ice, their interpretation with respect to individual deformation features is not straightforward (Bouchat & Tremblay, 2017; Hutter et al., 2018).

P2L29: one of which
Changed accordingly.

P2L34: rephrase
Changed to:
In addition, we test which conclusions about the properties of LKFs can be drawn from a spatio-temporal scaling analysis of sea ice deformation (following, e.g. Rampal et al., 2016; Hutter et al., 2018).

P3L2: outline?
Changed accordingly.

P3L28: Some have argued that power law is in the forcing? How sensitive are your results to the spatio-temporal length scales of the atmo/ocean forcing?
To our knowledge, only Hutter (2015) studied the impact of the wind forcing resolution on the scaling characteristics of sea ice deformation in idealized experiments. He found that increasing the horizontal grid spacing steepens the power-law of the spatial scaling, but also for coarse resolution forcing one can observe power-law scaling (see Spreen et al. 2017 for JRA-25 forcing with 1.125° resolution). So the magnitude of power-law exponent presented in our study might change slightly if a different wind forcing is used, however, the overall conclusion that the model reproduces the multi-fractal scaling will remain.

P4L8: we branch -> meaning?
Rephrased to:
On October 17th, 1995 the simulation with an ice thickness distribution (Thorndike et al., 1975) with 5 thickness categories separated by boundaries at 0.0m, 0.64m, 1.39m, 2.47m, and 4.57m is started.

P4L13: justify this choice. Cite Landy et al, 2019
There is extensive literature that ITDs observed in the field are characterized by negative exponential or lognormal distributions. The study mentioned by the reviewer uses these distributions to simulate SAR altimeter echos. In the revised manuscript, we give reference to a dedicated studies on ITDs and a review chapter. The mode of ⅔ of lognormal distribution was chosen by visually comparing observed ITDs. We

note here that these distributions are only the initial condition and will evolve during the spin-up. The text is changed:

Therefore, we use the fact that observed ITDs follow log-normal functions (Wadhams,1992; Haas, 2010) and describe the ITD of each grid-cell by a log-normal distribution with a mode of 2/3 of the mean thickness.

P4L19: bold not a good idea. I suggest run_ITD run_noITD
We agree and replace bold face by quotation marks: "ITD", ''noITD".

P4L32: what boundary?
The deformation rates that are calculated with the line integral approximation depend on the boundary of the cell along which the integration takes place. Lindsay and Stern (2003) showed that for strongly deformed cells the computed deformation rates depend on the number of vertices, which define the boundary of the cell. They refer to this uncertainty in deformation rates, introduced to the RGPS data set by only 4 vertices per cell, as boundary definition error. We changed the text to make clear that this uncertainty is present in strongly deformed cells:

For an accurate magnitude of the deformation rates and in particular the temporal scaling, only the most sophisticated option (1) can be used as it takes the advection of ice into account and addresses the effect of distorted vertices on the computation of the deformation rates (Lindsay & Stern, 2003) consistently for model and RGPS.

P5L9: contradicts power law distribution and localisation
This sentence refers to the parameters of the detection algorithm. As the detection algorithm works in pixel space, all parameters given in pixel units need to be adjusted to different resolution of the input data. We do not see how this contradicts the power-law scaling as the resolution used is always the lower bound of scaling behaviour.

P5L15: and Weiss et al, 2018 for space-time power laws
For clarity we refrain to state explicitly the space-time power-laws as a separate equation but refer to the paper:
Sea-ice deformation is known to depend on spatial and temporal scales following a power-law (Weiss, 2013; Weiss & Dansereau, 2017, for spaced-time coupled form), ...

P5L29: this is not clear here and some repeat of earlier paper might be needed
We changed to text to be more specific:
The spatio-temporal scaling analysis performed in this paper is based on Lagrangian drift data as suggested in Section 2.3.2. To transfer the RGPS sampling to the model output, we convert the regular gridded velocity output of the model to Lagrangian drift data by integrating trajectories from daily averaged velocity output of the model. Virtual buoys are initialized on the RGPS grid on November 1st of each year (1996-2007). The virtual buoys are advected with the modeled ice drift until mid-May of the following year and their positions are recorded every day.

P6L9: than ...(L<L0/2)
Changed accordingly.

P6L19: unclear sentences. What two streams are you referring to in this sentence?

The RGPS Lagrangian data sets is distributed in so-called streams. Each stream refers to one specific constellation of two overfly paths of RADARSAT that overlap in the observed region with a time difference of roughly 3 days. The formulation in the manuscript, therefore, was misleading and changed to:
Note that in this way the positions of drifters that are on both images are updated twice within a time period much shorter than 3 days. The time difference within one overfly (order of minutes) is small compared to the time difference between two different overflies that cover the same region (order of 3days).

P6L33: Brief algorithm schematic needed in appendix or clear reference to previous paper, section etc. . .
We included a reference to the Data User's Handbook of RGPS (Kwok & Cunningham, 2014), where detailed information about streams and deformation rate computation by with line integrals can be found.
For a more accurate conversion, we take the following processing steps for each RGPS stream (all points that are covered by two consecutive overflies of the satellite, for details see Kwok and Cunningham, 2014): ….

P7L4: General comment: it would be good to know what tuning you have undergone to achieve such a good
fit with the observations.
We perform no tuning, please see answer to major comment for details.

P7L5: decreases
Changed accordingly.

P7L16: Can you also check the space-time scaling as discussed in Weiss et al, 2018
With our method we can check the dependence of the spatial scaling exponent to temporal scales and vice versa. With this it is possible to test for space-time coupling as discussed in Weiss & Dansereau (2017). The space-time coupling discussed in Weiss & Dansereau (2017) and also in the original paper Marsan & Weiss (2010) is based on buoy data, from which a proxy for deformation is derived (Rampal et al., 2008). These observations have a coarse resolution, but cover a long time record. For the high-resolution RGPS data-set, we find that this space-time coupling varies strongly from year to year, with years showing the coupling and years not showing the coupling. Averaging over the entire RGPS period, we do not observe clear space-time coupling (to our knowledge also no other study shows the space-time coupling for RGPS). Therefore, we omitted to discuss this coupling in the manuscript and decided to show the multi-fractal properties as discussed in most recent modelling scaling studies (Rampal et al., 2019).

P8L11: Not clear if it is not good in this study or in Rampal's. Rephrase
The quality of the power-law fit in our study is comparable to the one in Rampal et al. (2019). For clarity, we therefore removed the statement and the sentence reads now as follows:
The curvature of the structure function of the temporal and spatial scaling exponent follows a power-law (Fig. 3 a and b) as suggested by Rampal et al. (2019).

P8L16: how does this link with power law exponents? Explain
We added a sentence to make this clear:
Due to the reduced ice strength, deformation increases yielding to a stronger localization of deformation in space and time and thereby higher scaling exponents.

P9L2: This is slightly too strong as they were developed also to represent some physical characteristics (i.e. stress redistribution...)

We reformulated the statement to be more general:

In summary, the spatio-temporal scaling analysis shows that both model simulations reproduce the observed multi-fractal heterogeneity and intermittency of sea-ice deformation (Marsan et al., 2004; Rampal et al., 2008; Weiss & Dansereau, 2017; Oikkonen et al., 2017) equally well as more sophisticated models that were specifically designed with these characteristics in mind (Girard et al., 2011).

P10L1:Important consideration is how the results presented below scale with model resolution but also with spatio-temporal scales of the forcing fields.

We agree that a discussion of the effect of both model and forcing resolution on the presented statistics is of interest for the reader. As an additional sensitivity study is out of the scope of this manuscript, we add the paragraph already stated in our answer to major comment #3 to the Discussion Section.

P10L9: does not seem significant and also raises questions as to how LKFs are detected in a changing Arctic

Changed to:

The RGPS LKF data-set shows little variation in the number of deformation features in the entire observing period with no clear trend (from 0.015 to 0.0125 LKFs per RGPS observation).

P10L22: any suggestions as to why? Generally little elements of physical explanations of the results are given.

We attribute the lower numbers of LKFs in the noITD simulation to the lack of inhomogeneities in the ice that facilitate sea ice deformation. For the ITD simulation, the different ice strength parameterisation (Rothrock, 1975) introduces inhomogeneities in the ice strength. We already discuss this link in the Discussions. We speculate that the higher seasonal variability in the ITD simulation is caused by a higher sensitivity to variations in the atmospheric forcing (such as storms) due the introduced inhomogeneities.

P10L33: Not clear to me how this relative density is calculated (what unit?) and how you can compare it to MODIS or CS2 information. For CS2 please also cite recent paper by Horvat et al, 2019.

We subdivide the Arctic Ocean in 50x50km boxes and count how many times we find LKF pixels in this box. This frequency is then normalized with the number of RGPS observations within the box, to compensate for the varying coverage of RGPS. To be more clear, we added more information to the caption of Figure 5:

Figure 5. (a,c,e) The density of LKFs in the RGPS data set and the two model simulations for the winters between 1996 and 2008 computed in 50x50km boxes over the Arctic Ocean. The absolute frequency of LKF pixels in a box is normalized by the total number of pixels with deformation in the data set. Only boxes with more than 500 deformation pixels in space and time are shown.

Therefore, the computed frequencies can be compared to lead frequencies derived from MODIS and CryoSat-2 that basically determine the frequency of open water pixels. However, the LKF densities in our study also include pressure ridges, which needs to take into account once comparing both.

Horvat et al. (2019) studies the floe size distribution from CryoSat-2. The number of floes and the spacing between them is needed to directly infer lead densities from the floe size distribution. Since the latter is not given in Horvat et al. (2019) no direct inferences on the lead density is possible, such that we decided not to include this study in our paper.

P13L17: Another possibility is that some of these features are ocean driven (geostrophic current or Eddies). There is extensive recent literature on this in this region,

In the manuscript we already name the Beaufort Gyre as one ocean driver of sea ice deformation in the Beaufort Sea. We agree that eddies generated within the gyre might also play a role in exerting stress on sea ice on small spatial scales. We included this to the text:

We do not observe the increased probability in LKF formation in either simulations, which may suggest that the Beaufort Gyre circulation is too weak (Willmes & Heinemann, 2016), there are too few mesoscale eddies (Zhao et al., 2014), or that the ice-ocean drag parameterization that does not take into account keels and sails in deformed multi-year ice is too simple (Tsamados et al., 2014; Castellani et al., 2018).

P14 Figure 6: over what period? Season? Specify in caption

The period of all statistics presented in the manuscript is the entire RGPS period (winters 1996 to 2008). We added this to the figure caption to be more clear:

Figure 6. (a,b,c) Mean orientation of LKFs for RGPS and two model simulations for the winters between 1996 and 2008. ...

P15L7: good review

Thank you.

P15L12: So not clear what method you use to measure LKF lengths

The length of the LKF is given by summing the distances between the individual pixels of the LKF. We added this information to the manuscript:

We determine the PDF of LKF lengths from RGPS data and both model simulations (Fig. 7a). The LKF length is measured as the cumulative sum of the distance between pixels along the LKF.

P16L29: Cite also study by Hibler and Hutchings, 2004, + several studies by Wilchinsky + Feltham + Tsamados + Heorton on anisotropic rheology with prescribed diamond shaped floes. Tsamados et al, 2013 describes sensitivity to this intersection angle See also papers by Cunningham et al, 1994, Schulson et al, 2006, but also Gray, Coon, Pritchard, Maslowski, Ukita, Moritz...

We acknowledge that there are plenty of studies dealing with intersection angles of deformation lines and thank the reviewer to highlight a few. Given that the referred section deals with the disagreement of modeled and observed intersection angles, there are three different categories of papers that we would like to cite: (i) studies of observed intersection angles intersection angles, (ii) studies that link the intersection angle to the material properties of the ice (namely the rheology), (iii) studies that modeled deformation lines and compared intersection angles. We added for (i) Cunningham et al. (1994) and Schulson et al. (2006), for (ii) Utika and Moritz (1995) (we already cited the suggested Pritchard, 1988), and for (iii) Hutchings et al. (2005).

The papers on the anisotropic rheology deal with the parameterisation of given intersection angles on subgrid scale. In the paper, however, we study the intersection of grid-scale LKFs simulated by the model. We do not see how both can be related despite the high quality of named studies. Therefore, we refrain from citing them in this part of the manuscript.

The paragraph now reads as follows:

The PDF of intersection angles for RGPS data peaks around 40°-50° (Fig. 9). This peak agrees with typical intersection angles of 30°-50° inferred from satellite imagery (Walter and Overland, 1993; Cunningham et al., 1994; Schulson, 2004; Wang, 2007) and laboratory measurements (Schulson et al.,

2006). We find the lowest probabilities for angels smaller than 20°. Angles larger than 50° occur more often than angles smaller than 40°. The distributions of intersection angles in both model simulations are very different from the RGPS data and peak at 90°, which is in agreement with idealized experiments using the VP rheology (Hutchings et al., 2005). Intersection angles smaller than 60° are less frequent in the model simulations than in the RGPS data. The differences between both simulations are small. According to theoretical considerations, the intersection angle is determined by the slope of the yield curve (Pritchard, 1988; Ukita and Moritz, 1995; Wang, 2007). As both simulations use the same elliptical yield curve with a normal flow-rule (Hibler, 1979) similar intersection angles of LKFs are expected. We attribute the small differences in Fig. 9 to sea ice fields with a different amount of LKFs. Ringeisen et al. (2019) derived for idealized compression experiments that it is impossible to obtain intersection angles smaller than 60° with an elliptical yield curve. This explains the deficit of small intersection angles in our simulations.

P17 figure 9: So quite important structural difference of the model with reality here.
Therefore we summarize: "*Although the model reproduces most LKF statistics, it completely fails to simulate the observed distribution of LKF intersection angles.*" in the Discussions.

P18L5: angles
Changed accordingly.

P18L17: define lifetime calculation (algo).
We added information to make it clearer:
We determine the lifetime of an LKF by counting how many times we track a feature. The lifetime estimates are binned into 3-day intervals, that is, the temporal resolution of the deformation data. If an LKF can not be tracked, we assign it to the lowest lifetime class (0-3 days). Tracked LKFs are assigned to a lifetime class according to the number of tracks (one time tracked is assigned to 3-6 days, two time tracked to 6-9 days, etc.).
How model resolution is this?
As discussed already in the answer to the major comment, the temporal resolution of 120s is much smaller than the smallest lifetime regarded (0-3 days). Thus we do not expect that the presented PDF of lifetimes is impacted by the choice of temporal resolution.
Do you calculate persistence in a lagrangian or eulerian way?
As we track feature in time using the drift information, our persistence can be regarded as Lagrangian quantity.

P18L32: Why didn't you estimate similar biases for the other LKFs characteristics discussed earlier?
The bias the reviewer refers to is an uncertainty in the tracked features caused by the temporal varying spatial coverage of the RGPS data-set. This primarily impacts all LKF characteristics that are based on the result of the tracking algorithm, that are, LKF persistence and growth rates. For both quantities, we already discuss the effects of this uncertainty.

P22L19: or indirectly in the anisotropic rheologies of Tsamados et al, 2013 via the additional dynamics on the order parameter controlling the degree of anisotropy
We added:

Note that the local degree of anisotropy of the elastic-anisotropic plastic rheology (Tsamados et al. 2013) also represents a memory of past deformation.

P22L32: see also Heorton et al, 2019
Please see comment above for new literature in the section of intersection angles.

**References:**

Cunningham, G. F., Kwok, R., and Banfield, J.: Ice lead orientation characteristics in the winter Beaufort Sea, in: Proceedings of IGARSS '94 - 1994 IEEE International Geoscience and Remote Sensing Symposium, vol. 3, pp. 1747–1749 vol.3, https://doi.org/10.1109/IGARSS.1994.399553, 1994.

Haas, C.: Dynamics Versus Thermodynamics: The Sea Ice Thickness Distribution, chap. 4, pp. 113–151, John Wiley Sons, Ltd, https://doi.org/10.1002/9781444317145.ch4, https://onlinelibrary.wiley.com/doi/abs/10.1002/9781444317145.ch4, 2010.

Heorton H. D. B. S., Feltham D. L. and Tsamados M., Stress and deformation characteristics of sea ice in a high-resolution, anisotropic sea ice model; *Philosophical Transactions of the Royal Society A: Mathematical, Physical and Engineering Sciences;* http://doi.org/10.1098/rsta.2017.0349; 2018

Horvat, C., E. Tziperman, and J.-M. Campin (2016), Interaction of sea ice floe size, ocean eddies, and sea ice melting, Geophys. Res. Lett., 43, 8083–8090, doi:10.1002/2016GL069742.

Horvat, C., Roach, L., Tilling, R., Bitz, C., Fox-Kemper, B., Guider, C., Hill, K., Ridout, A., and Sheperd, A.: Estimating The Sea Ice Floe Size Distribution Using Satellite Altimetry: Theory, Climatology, and Model Comparison, The Cryosphere Discuss., https://doi.org/10.5194/tc-2019-134, in review, 2019.

Hutchings, J. K., Heil, P., and Hibler, W. D.: Modeling Linear Kinematic Features in Sea Ice, Monthly Weather Review, 133, 3481–3497, https://doi.org/10.1175/MWR3045.1, https://doi.org/10.1175/MWR3045.1, 2005.

Hutter, N.: Viscous-plastic sea-ice models at very high resolution, Master's thesis, University of Bremen, Alfred Wegener Institute, Helmholtz Centre for Polar and Marine research, https://doi.org/10013/epic.46129, http://dx.doi.org/10013/epic.46129, 2015

Koldunov, N. V., Danilov, S., Sidorenko, D., Hutter, N., Losch, M., Goessling, H., et al. (2019). Fast EVP solutions in a high-resolution sea ice model. *Journal of Advances in Modeling Earth Systems*, 11, 1269– 1284. https://doi.org/10.1029/2018MS001485

Kwok, R. and Cunningham, G. F.: RADARSAT GEOPHYSICAL PROCESSOR SYSTEM: DATA USER'S HANDBOOK (Version 2.0), 2014.

Levy, G., Coon, M., Nguyen, G., and Sulsky, D.: Metrics for evaluating linear features, Geophysical Research Letters, 35, https://doi.org/10.1029/2008GL035086, https://agupubs.onlinelibrary.wiley.com/doi/abs/10.1029/2008GL035086, 2008.

Lüpkes, C. and Gryanik, V. (2015): Parameterization of drag coefficients over polar sea ice for climate models, Mercator Ocean Quarterly Newsletter - Special Issue, 51 , pp. 29-34.

Manucharyan, G. E, & Thompson, A. F. (2017).  Submesoscale sea ice-ocean interactions in marginal ice zones. *Journal of Geophysical Research: Oceans*,  122,  9455– 9475. https://doi.org/10.1002/2017JC012895

Menemenlis, D., I. Fukumori, and T. Lee (2005), Using Green's functions to calibrate an ocean general circulation model, Mon. Weather Rev., 133(5), 1224–1240, doi:10.1175/MWR2912.1.

Mohammadi-Aragh, M., Goessling, H. F., Losch, M., Hutter, N., and Jung, T.: Predictability of Arctic sea ice on weather time scales, Scientific reports, 8, 2018.

Schulson, E., Fortt, A., Iliescu, D., and Renshaw, C.: On the role of frictional sliding in the compressive fracture of ice and granite: Terminal vs. post-terminal failure, Acta Materialia, 54, 3923 – 3932, https://doi.org/https://doi.org/10.1016/j.actamat.2006.04.024, http://www.sciencedirect.com/science/article/pii/S1359645406003120, 2006.

Sumata, H., F. Kauker, M. Karcher, and R. Gerdes, 2019: Simultaneous Parameter Optimization of an Arctic Sea Ice–Ocean Model by a Genetic Algorithm. Mon. Wea. Rev.,147, 1899–1926, https://doi.org/10.1175/MWR-D-18-0360.1
Ukita, J., and R. Moritz (1994), Yield curves and flow rules of pack ice, Journal of Geophysical Research-Oceans, 100(C3), 4545–4557.

Tsamados, M., Feltham, D. L., and Wilchinsky, A. V.: Impact of a new anisotropic rheology on simulations of Arctic sea ice, Journal of Geophysical Research: Oceans, 118, 91–107, https://doi.org/10.1029/2012JC007990, https://agupubs.onlinelibrary.wiley.com/doi/abs/10.1029/2012JC007990, 2013.

Ukita, J. and Moritz, R. E.: Yield curves and flow rules of pack ice, Journal of Geophysical Research: Oceans, 100, 4545–4557, https://doi.org/10.1029/94JC02202, https://agupubs.onlinelibrary.wiley.com/doi/abs/10.1029/94JC02202, 1995

Wadhams, P.: Sea ice thickness distribution in the Greenland Sea and Eurasian Basin, May 1987, Journal of Geophysical Research: Oceans, 97, 5331–5348, https://doi.org/10.1029/91JC03137, https://agupubs.onlinelibrary.wiley.com/doi/abs/10.1029/91JC03137, 1992.

Zhao, M., Timmermans, M.-L., Cole, S., Krishfield, R., Proshutinsky, A., and Toole, J.: Characterizing the eddy field in the Arctic Ocean halo- cline, Journal of Geophysical Research: Oceans, 119, 8800–8817, https://doi.org/10.1002/2014JC010488, https://agupubs.onlinelibrary.wiley.com/doi/abs/10.1002/2014JC010488, 2014.

---

## Author Response (AR1)

**Answers to the editor**

Dear Mr. Hutter and Dr. Losch,

Thank you for your details response to the reviewers. I have some additional comments to add after consideration of the reviews and your response. Thank you to the reviewers for identifying points that were not clear in the paper.

I trust you will ensure that the paper is clear especially in regard to its goals and your contribution towards improving assessment of sea ice deformation in models. I am in agreement with you that more metrics, other than spatial and temporal scaling, are needed. It is an important point that scaling relationships are not a unique indicator that a simulated deformation field is physically realistic.

I believe it is important to keep in mind the limitations of the viscous-plastic model. I found that instabilities in the stress-velocity field can follow model grid lines. This might be one reason you find modeled LKF intersection angles cluster around 90 degrees. While I agree that a full sensitivity study is beyond the scope of this paper, such studies have been performed with more idealised models and could help in understanding the persistence and density behaviour of the viscous-plastic model. The fact that there is an increase in instability density with the ITD is not surprising, but does need to be pointed out to people who use these models. As you point out the ITD allows steep gradients in the stress field (ice strength being a function of ice thickness), that may not present with a smoother thickness field. It is very important, from my experience, in idealised sensitivity studies to control for various sources of stress discontinuities: from poorly filtered wind fields due to interpolation procedure, to poorly converging viscous-plastic solution, to the parameterisation of the strength.

Thank you for highlighting these points. Following your observations that instabilities follow model grid lines, we compared the orientation of the modeled LKFs with the orientation of grid lines (Fig. 1 of this answer). There are regions where one mode of the LKF orientation aligns with the grid lines, but we do not observe this behaviour consistently in the entire Arctic. In addition, small LKFs are most likely to follow the grid orientation. For this reason, we limit the analysis to LKFs that are larger than 10 pixel (=125km). We added a sentence to the section of intersection angles and indicate the orientation of the numerical grid in Fig. 6 (showing the LKF orientation):

The peak in the PDF near 90° suggests a dominant LKF alignment with the numerical grid. A close inspection, however, does not show this dominant alignment (Fig. 6b,c,e,f). Therefore, we can assume that the grid orientation has only a small effect on the LKF orientation, but that the rheology itself causes the overestimation of the intersection angle.

(e) MITgcm 2km

[Figure]

**Figure 1:** Modes of the LKF orientation in the 2km MITgcm simulation without ITD. Same figure as subfigure (e) in Figure 6 of the manuscript, but including every 100th grid line of model grid in red dashed lines.

We agree that it is important to highlight the increased amount of instabilities in the ITD simulation (we call them inhomogeneities in the manuscript) that initiates LKF formation to the reader. We already do so in the Introduction:

With our analysis we test whether the ice strength parameterization of the ITD model, which mainly depends on the thinner ice classes, accelerates lead formation by a faster feedback between deformation, ice thickness, and ice strength as suggested in Hutter et al. (2018).

and in the Discussion section:

With an ITD, the number and density of LKFs increase significantly. In the ITD simulation shear and divergence have a strong impact on the thin thickness classes which immediately feeds back into the ice strength facilitating further deformation. Therefore, inhomogeneities introduced by deformation in the thickness fields are much stronger compared to the standard VP simulation.

We agree that idealized experiments have a great potential to disentangle the effects of the various drivers of sea ice deformation. We already reference a number of idealized experiments (Hutchings et al., 2005; Hutter, 2015; Dansereau et al. 2016; Heorton et al. 2018, Ringeisen et al., 2019) and discuss how they help to understand the presented results from our Pan-Arctic simulations. In addition, we now encourage the use of idealized experiments in the Discussions Section:

We suggest to disentangle the effects of model resolution and wind forcing, but also ice strength parameterisation and solver parameters, on the formation of LKFs in a sensitivity study. For such a study, idealized experiments appear most suitable as they easily allow for higher number of simulations and to isolate effects.

I would like to point out an incorrect definition that is in the title of your paper. Two kilometers is not lead resolving. True, some leads may be two kilometers or more in width, but there is a distribution of lead and crack widths from sub-meter to many kilometers. You need to be careful about this, in the title to your paper and elsewhere where you state the models are lead resolving.

We agree that the 2-km simulations in the manuscript do not resolve the entire spectrum of lead width. Nevertheless, simulations at these high resolution are unique in a way that leads (although wide ones) are simulated as features of reduced ice thickness and concentration as well as localized deformation. In analogy to simulating eddies in ocean models, we now describe our simulations as lead-permitting in a sense that the simulations resolves leads with a width larger than the grid resolution. The new title reads as,

Feature-based comparison of sea-ice deformation in lead-permitting sea-ice simulations

and we replaced "lead-resolving" by "lead-permitting" in the entire manuscript.

While I agree with much of your response to the reviewers there are some points I feel you should address.

I also on my first read of the paper questions if the two simulations had been cherry picked. Your description of how the runs were set up was very helpful and should be included in the paper.

We added a brief description of the model performance to the text, as already suggested in the answers to the reviewers. In addition, we decided to add the full description and an additional figure that shows time series of the modeled sea ice volume and extent (same as in the answers to the reviewers) as an appendix to the paper.

Regarding your upper limit in angle between sections that are pieced together into LKFs, have you considered the high curvature tidal cracking can form? I realise this is outside of the scope of your paper, but I do caution you to consider that this impact of the cut off will change with model resolution and also

the kinematics a model can resolve. Does your cut off length of 125km preclude inclusion of features with high curvature?

The detection algorithm has limitations in detecting short high-curvature leads. Larger LKFs can show higher curvatures, which we also observe as wider spread in curvature in Fig. 8 of the manuscript. The parameters of the algorithms (also the accepted difference in angle that weighs the impact of the difference in orientation of two segments in the reconnection instance) are optimized to fit the majority of LKFs. As most of them are quasi-linear with little curvature, the small number of short high-curvature leads might not be picked up by the algorithm with the same reliability. Nevertheless, Linow & Dierking (2017) studied the curvature of LKFs from hand-picked data and automatic detected LKFs and found low curvatures with close to linear LKFs in both cases. Thus, we are confident to describe the LKF shapes as scale invariant, but add an additional sentence to the LKF curvature section:

The spread in curvature decreases at smaller LKF length (<200km) for RGPS and both simulations, because the LKF detection algorithm does a poor job of detecting short high-curvature LKFs.

The limit of 125km is only used to determine the intersection angle of LKFs. The curvature analysis runs on all detected LKFs and thus has a lower limit for LKF lengths of 37.5km. We could limit the curvature analysis to LKFs larger than 200km to preclude the underestimation of short high-curvature LKFs by the detection algorithm. However, we prefer to show Figure 8 as is and comment on this limitation given the small amount of these features at the resolution used in our study.

Some specific comments:

page 16, line 27: constraining your analysis to LKFs that form in the same time record may not be a strict enough constraint to ensure LKFs formed at the same time. RGPS images can be days apart. While this is the best you can do, I feel you need to acknowledge the limitation.

We agree and added:

Therefore, we limit the analysis of intersection angles to pairs of LKFs that form in the same time record. **We note that with this restriction the maximum time between the formation of both LKFs is determined by the temporal resolution of RGPS of 3days. Therefore some LKF pairs may not have formed simultaneously.**

If the mechanism creating LKFs in the viscous-plastic model is an instability in the model itself, how does this compare to actual quasi-brittle or granular failure? I ask this question because I feel it is important that we understand how this instability is controlled by noise in the ice strength (or stress) field and I wonder if we can parameterise this to produce realistics LKF densities. The use of a damage parameter and brittle failure is perhaps shifting this to a more physically tractable representation, yet you may still need to control for ice age in some way in what ever parameterisation you use.

The plastic failure that creates LKFs in the high resolution VP simulations is similar to brittle failure with the difference that the information of past deformation is stored in the ice thickness and concentration and not in a separate damage parameter. By varying the dependence of the ice strength on the ice thickness and concentration, we can amplify the effect of inhomogeneities in the thickness and concentration fields on the ice strength and triggering failure. In the ITD simulation exactly this is happening with an increasing number of LKFs and a realistic LKF density. However, we observe that the model needs too much time to close a lead and level out the associated inhomogeneities. The benefit of an additional damage parameter it that it allows to adjust this closing rate with a healing parameter separate from the thermodynamic processes associated with ice growth.

Finally, density and number is an interesting parameter to compare between LKFs extracted from 10km (roughly) resolution RGPS and 2km resolution model. Perhaps I missed this, but how did you control for this?
Due to the Lagrangian nature of the RGPS data set, the RGPS deformation rates cumulate the deformation occurring all LKFs within a grid box. Thus the RGPS deformation rates also include imprints of LKFs at smaller scales than the grid resolution of 10km. In the model, multiple grid points are needed to resolve a lead, so that the nominal resolution for leads is lower than 2km. To compare the density and numbers of LKFs on grids with different resolution, we normalize these quantities according to number of observations and horizontal grid spacing. The parameters of the detection algorithm that depend on the horizontal grid spacing are also adjusted accordingly.

Best regards,
Jenny